# Dependence on Myb expression is attenuated in myeloid leukaemia with N-terminal CEBPA mutations

Giacomo Volpe[1,2] ⓘ, Pierre Cauchy[3], David S Walton[1], Carl Ward[1,2] ⓘ, Daniel Blakemore[1], Rachael Bayley[1], Mary L Clarke[1], Luisa Schmidt[4] ⓘ, Claus Nerlov[5], Paloma Garcia[1] ⓘ, Stéphanie Dumon[1], Florian Grebien[4,6] ⓘ, Jon Frampton[1] ⓘ

**Mutations at the N- or C-terminus of C/EBPα are frequent in acute myeloid leukaemia (AML) with normal karyotype. Here, we investigate the role of the transcription factor Myb in AMLs driven by different combinations of CEBPA mutations. Using knockdown of Myb in murine cell lines modelling the spectrum of CEBPA mutations, we show that the effect of reduced Myb depends on the mutational status of the two Cebpa alleles. Importantly, Myb knockdown fails to override the block in myeloid differentiation in cells with biallelic N-terminal C/EBPα mutations, demonstrating for the first time that the dependency on Myb is much lower in AML with this mutational profile. By comparing gene expression following Myb knockdown and chromatin immunoprecipitation sequencing data for the binding of C/EBPα isoforms, we provide evidence for a functional cooperation between C/EBPα and Myb in the maintenance of AML. This co-dependency breaks down when both alleles of CEBPA harbour N-terminal mutations, as a subset of C/EBPα-regulated genes only bind the short p30 C/EBPα isoform and, unlike other C/EBPα-regulated genes, do so without a requirement for Myb.**

## Introduction

Acute myeloid leukaemia (AML), one of the most common and deadliest forms of proliferative neoplasms, is established through a stepwise acquisition of genetic and epigenetic alterations that result in the malignant transformation of haematopoietic progenitor cells (Kelly & Gilliland, 2002; Moore, 2005). Often, AML arises through the collaboration between mutations affecting transcription factors (e.g., CEBPA, PU.1, and RUNX1) and signalling proteins (such as FLT3, RAS, and KIT) that lead to an aberrant proliferation capacity coupled with a disruption of terminal myeloid differentiation (Tenen, 2003; Rosenbauer & Tenen, 2007).

C/EBPα, a leucine zipper transcription factor with a known tumour suppressor function, has been demonstrated to play an important role in granulocytic development and in the maintenance of haematopoietic stem cell homeostasis (Porse et al, 2001, 2005; Zhang et al, 2004; Koschmieder et al, 2009; Welner et al, 2013; Ye et al, 2013). C/EBPα is translated as two major isoforms, namely a full-length 42-kD form (p42) and a truncated 30-kD protein (p30) that arises from a downstream translational initiation codon (Lin et al, 1993). Mutations in the CEBPA gene are frequently associated with leukaemia, being found in 8–14% of all de novo AML with normal karyotype (Nerlov, 2004; Leroy et al, 2005; Song et al, 2015) and typically involve both alleles. C/EBPα-mutant proteins are classified into two major groups: (i) C-terminal insertions or deletions within the basic region leucine zipper DNA-binding domain; and (ii) N-terminal mutations that lead to the complete ablation of p42 while retaining normal p30 function (Pabst et al, 2001; Leroy et al, 2005; Fasan et al, 2014). Most patients carrying CEBPA mutations harbour one allele with an N-terminal mutation and one with a C-terminal mutation, with homozygosity for N- or C-terminal mutations being less common (Gombart et al, 2002; Pabst & Mueller, 2007). Furthermore, several reports have demonstrated that biallelic mutations of CEBPA are associated with a favourable outcome, when not found in association with FLT3-activating mutations (Renneville et al, 2009; Dufour et al, 2010).

Efforts aimed at understanding how mutations or oncoproteins may cooperate in driving the leukaemogenesis have pointed to cooperation between C/EBPα and other transcription factors, such as RUNX1, MYB, and PU.1. We have previously demonstrated the functional cooperation of Myb and C/EBPα in the regulation of the Flt3 gene in both haematopoietic and leukaemia stem cells (Volpe et al, 2013, 2015). Our studies indicated that Myb and C/EBPα act cooperatively through their combined activity on promoter and intronic elements in the Flt3 gene (Volpe et al, 2013). Furthermore, we reported a strong linear correlation between expression of the

[1]Institute of Cancer and Genomic Sciences, College of Medical and Dental Sciences, University of Birmingham, Birmingham, UK  [2]Key Laboratory of Regenerative Biology, Joint School of Life Sciences, Guangzhou Institutes of Biomedicine and Health, Chinese Academy of Sciences, Guangzhou, and Guangzhou Medical University, Guangzhou, China  [3]Max Planck Institute of Immunobiology and Epigenetics, Freiburg, Germany  [4]Ludwig Boltzmann Institute for Cancer Research, Vienna, Austria  [5]Medical Research Council Molecular Haematology Unit, Weatherall Institute of Molecular Medicine, University of Oxford, Oxford, UK  [6]Institute of Medical Biochemistry, University of Veterinary Medicine, Vienna, Austria

Correspondence: j.frampton@bham.ac.uk; g.volpe@bham.ac.uk
Giacomo Volpe and Pierre Cauchy are joint first authors
Stéphanie Dumon, Florian Grebien, and Jon Frampton are joint senior authors

two transcription factors and *FLT3* RNA levels in human CN-AML, adding to an increasing body of evidence that points to MYB being a crucial component of leukaemia maintenance and oncogene addiction (Hess et al, 2006; Zuber et al, 2011; Clarke et al, 2017).

Our findings on the cooperation of Myb and C/EBPα in *Flt3* gene regulation prompted us to investigate the global extent of this cooperation in leukaemia and to determine how manipulation of Myb expression might impact on the maintenance of C/EBPα-driven leukaemia. To address this, we performed genetic manipulation studies in murine haematopoietic progenitor cell lines harbouring either wild-type C/EBPα or the most frequently occurring combinations of biallelic CEBPA mutations, that is $N^{ter}/N^{ter}$ or $N^{ter}/C^{ter}$ to determine the biological and molecular consequences of reduced Myb activity on the leukaemia driven by those mutations. Here, we show that reducing Myb activity can override the differentiation barrier, although the dependency on *Myb* expression generally observed in leukaemia is minimal in the presence of CEBPA biallelic N-terminal mutations.

# Materials and Methods

### Cell lines

Cells were cultured in RPMI medium supplemented with 10% fetal bovine serum, 50U/ml penicillin, 50 μg/ml streptomycin, and 2 mM ʟ-glutamine. The culture of FMH9 cells (Volpe et al, 2013) were supplemented with 50 ng/ml stem cell factor, 5 ng/ml GM-CSF, 5 ng/ml IL-3 (IL3), and 5 ng IL-6 (IL6), whereas KL cells (GV, JF, and FG, unpublished) and LL cells (Grebien et al, 2015; Schmidt et al, 2019) required 2 ng/ml IL3. Both KL and LL have been established by serial replating of E14.5 foetal liver cells obtained from mice homozygous for the Lp30 allele (Kirstetter et al, 2008) or carrying both Lp30 and K313KK alleles (Bereshchenko et al, 2009). Briefly, the KL and LL cell lines were obtained by performing six rounds of replating in M3434 semisolid medium (Stem Cell Technologies Inc) followed by an initial liquid culture for 4 wk in the presence of stem cell factor (50 ng/ml), IL3 (2 ng/ml), and IL6 (2 ng/ml). After this initial period, the cells were transferred into the culture medium described above. All cytokines were purchased from Peprotech EC.

### Transfection experiments, cell viability, proliferation, apoptosis, and differentiation assays

In total, $5 \times 10^6$ FMH9, KL, or LL cells were electroporated with 300 mM of *Myb* siRNA (s70212, Ambion; Life Technologies) or a scrambled negative control siRNA (4390843 Silencer Select Negative Control #1; Life Technologies) using an Amaxa 4D-nucleofector with solution SF Cell Line (V4XC-2024; Lonza) and program EO-100 for FMH9 cells or solution P3 Primary Cells (V4XP-3024; Lonza) and program DS-120 for KL and LL cells.

After transfection, the cells were plated at a density of $10^6$ cells/ml and viable cells counted and passaged at a ratio of 1:2 every 24 h for 4 consecutive days. Cell cycle analysis was performed by labelling transfected cells (48 h post-nucleofection) with 10 μM BrdU for 1 h. Cells were co-stained with 7-AAD (A9400-1mg; Sigma-Aldrich) and BrdU using the BrdU flow kit (8811-6600; BD Bioscience) according to the manufacturer's instructions, as previously described (Bayley et al,

2018). Apoptosis analysis was performed using the Annexin V kit (eBioscience) as previously described (Volpe et al, 2015). The percentage of apoptotic cells was obtained by performing live cells gating. Proliferation analysis was performed using CellTrace carboxyfluorescein succinimidyl ester (CFSE) Cell Proliferation Kit (C34554; Thermofisher Scientific). Assessment of differentiation following *Myb* knockdown was achieved by flow cytometry/immunofluorescence staining of the cells with anti-CD11b PE-Cy7 (25-0112-81; eBioscience) anti-Gr-1 APC (14-5921-82; eBioscience), anti-CD135 PE (12-1351-81; eBioscience), and anti-CD117 PE-Cy5 (15-1171-82; eBioscience). Acquisition and analysis of flow cytometric data were performed using Cyan ADP with either Summit 4.4 software (Beckman Coulter) or FlowJo software (FlowJo, LLC).

### Quantitative reverse transcriptase polymerase chain reaction (RT-PCR) analysis

$10^6$ cells from each line were harvested 24 h post-transfection. RNA was extracted using RNeasy Mini kit (QIAGEN), and first-strand cDNA synthesis was performed using standard protocols. Quantitative PCR reactions were performed using predesigned Taqman gene expression assays as previously described (Volpe et al, 2015).

### Statistical analysis

Statistical significance was determined by performing *t* test for pairwise comparison, and the *P*-values are indicated where appropriate. Analysis of *MYB* expression in human patient array data presented in Fig 1A was performed using non-parametric Kruskal–Wallis test. All statistical analyses were performed using GraphPad Prism 7 (GraphPad Software Inc).

### Bioinformatic analyses

Details of the methodologies used to analyse and compare RNA-seq, microarray, and ChIP-seq data, including Gene Ontology (GO) and Gene Set Enrichment Analysis (GSEA) comparisons, can be found in the Supplementary Information.

### RNA-sequencing

For RNA-Seq, libraries were prepared using the Illumina TruSeq Stranded kit according to the manufacturer's instructions. Sequencing was performed at the Institute of Medical Biochemistry, University of Veterinary Medicine, Vienna, Austria, and in Genomics Birmingham, University of Birmingham, Birmingham, UK, on Illumina HiSeq 2500 and NextSeq 500 sequencers, respectively.

### Data availability

RNA-Seq data generated in this study are available at the Gene Expression Omnibus under series GSE119348.

# Results

### High *MYB* expression is associated with *CEBPA* mutations in AML

Previous studies have provided evidence for functional cooperation between C/EBPα and MYB in activating the expression of key genes

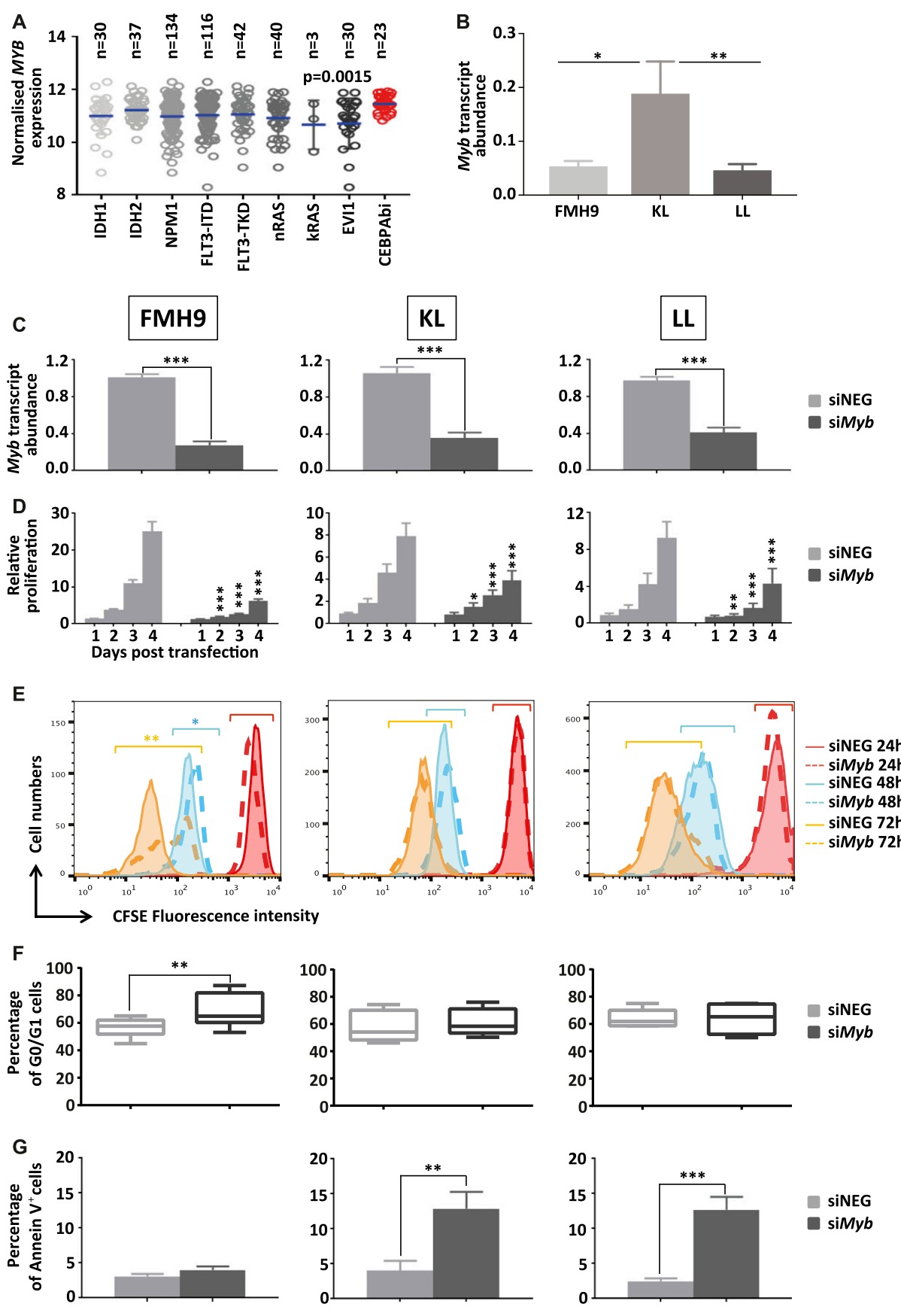

for both haematopoietic and leukaemia stem cell functions in mouse and human (Volpe et al, 2013, 2015, 2017). Using publicly available AML patient profiling arrays (Verhaak et al, 2009) and focussing on specific molecular abnormalities found in cytogenetically normal AML (CN-AML), we observed that *MYB* expression levels are highest in those patients carrying biallelic *CEBPA* mutations (Fig 1A).

Given the lack of a suitable cellular system to investigate the relationship between MYB and C/EBPα in human AML, we used murine cell lines modelling the spectrum of *CEBPA* mutations. Bereshchenko and coworkers demonstrated that mutations in the C/EBPα protein efficiently drive leukaemia in vivo and that the combination of N- and C-terminal mutations were the most highly leukaemogenic, whereas biallelic C-terminal mutation resulted in the longest latency (Kirstetter et al, 2008; Bereshchenko et al, 2009). We generated cell lines by performing serial replating in semisolid medium of E14.5 foetal liver cells carrying knock-in modifications mimicking either the N-terminal LP30 (L allele) or the C-terminal K313KK (K allele) *CEBPA* mutations (Grebien et al, 2015). LL cells were used to represent AML harbouring biallelic N-terminal *CEBPA* mutations, whereas cells carrying one K mutation and one L mutation (KL cells) provided a model of leukaemia with both N- and C-terminal *CEBPA* mutations. AML with wild-type C/EBPα expression was modelled using a previously characterised myelomonocytic leukaemia cell line, namely, FMH9 (Volpe et al, 2013), which was established by ectopic expression of HoxA9 and Meis1 in bone marrow haematopoietic progenitor cells. The phenotype of these cells lines was investigated by determining the surface expression of several myeloid markers (CD11b, Gr1, Kit, and Flt3) (Fig S1A). Importantly, mRNA quantification indicated that both FMH9 and LL cells displayed similar *Myb* mRNA levels, whereas KL cells exhibited a higher level of *Myb* expression (Fig 1B), thus being in agreement with the observations obtained from the patient array data.

### Manipulation of Myb expression does not reverse the differentiation block in cells carrying biallelic N-terminal *CEBPA* mutations

To investigate the requirement for Myb in the maintenance of CEBPA-driven leukaemia, we performed siRNA-mediated knock-down of *Myb*. Cells were transfected with siRNAs targeting either *Myb* or a scrambled negative control and were harvested after 24 h to determine the efficiency of knockdown. This analysis revealed a decrease in *Myb* transcripts by 60–80% in all cell lines (Fig 1C).

To determine the biological consequences of *Myb* knockdown cells were cultured for up to 96 h and cell numbers determined daily. *Myb* down-regulation induced growth retardation in the three cell lines, regardless of the *CEBPA* mutational status; albeit those cells showing a similar pattern, the growth defect observed in FMH9 cells was significantly more pronounced in comparison with the *CEBPA* mutant cells lines (Figs 1D and S1B). Flow cytometric analysis of CSFE dye dilution revealed that *Myb* knockdown induced a significant proliferation defect in FMH9 cells, whereas both KL and LL cells were unaffected (Figs 1E and S1C). The pattern of the proliferation defect observed in FMH9 cells 72 h post *Myb* down-regulation appeared to be bimodal, probably because of cells being induced to differentiate or due to the transient effect of the siRNA-mediated knockdown. By combined staining with 7-AAD and BRDU 48 h post-transfection, we observed that *Myb* knockdown led to a significant increase in the proportion of cells in the G0/G1 phase with a concomitant decrease in both S and G2/M phases in FMH9, whereas KL and LL cells showed no alteration in their capacity to progress through the cell cycle (Figs 1F and S2A, and B). This analysis also revealed an increase in the percentage of cells with less than 2n DNA content in *CEBPA*-mutant cell lines only, this being indicative of cells undergoing apoptosis/necrosis (Fig S2C). To confirm this observation, we performed Annexin V staining. This revealed a significant increase in the extent of apoptotic/necrotic cells in both KL and LL cells, whereas no increase was observed in FMH9 cells (Figs 1G and S3A).

Because it is accepted that homozygous *CEBPA* mutations lead to a block in myeloid lineage commitment, we investigated how *Myb* knockdown affects the differentiation capacity of cells in the presence of either wild-type or biallelic mutant C/EBPα. At 96 h, both FMH9 and KL cells exhibited a clear induction of myeloid differentiation as indicated by increased expression of Gr-1 and CD11b (Fig 2). However, this phenomenon was not observed in LL cells, which is intriguing because it suggests that leukaemia cells carrying biallelic N-terminal *CEBPA* mutations have a reduced dependency on *Myb* expression in respect to differentiation control.

### Molecular consequences of *Myb* manipulation in cell lines carrying either wild-type or mutant CEBPA

Myb was previously reported to suppress myeloid commitment and to promote self-renewal in haematopoietic progenitors and

**Figure 1. *Myb* expression is required for the proliferation of CEBPA biallelic mutant cell lines.**
**(A)** Scatter plot depicting the abundance of *MYB* transcript in subgroups of patients from the Verhaak et al (2009) dataset, characterised by the molecular abnormalities indicated in the graph. Statistical significance presented in this plot has been calculated using non-parametric the Kruskal–Wallis test. **(B)** Bar plot representing *Myb* mRNA quantification by quantitative RT-PCR in FMH9, KL, and LL cell lines, normalised against *B2m* house-keeping gene results. Statistical analysis was performed using t test (**P < 0.01 and *P < 0.05). **(C)** Quantitative RT-PCR of *Myb* transcript abundance in FMH9, KL, and LL cells 24 h post-transfection with *Myb* siRNA. Expression is normalised to *B2m* and standardised to the control samples. Error bars represent the SEM and numbers are plotted as mean ± SEM. Each plot is representative of six independent experiments (***P < 0.001 and *P < 0.05). **(D)** Bar plot indicating the cell viability and relative proliferation of FMH9, KL, and LL cells after *Myb* siRNA transfection relative to the corresponding negative control (***P < 0.001 and **P < 0.01, *P < 0.05). This plot represents an average of six independent experiments. **(E)** Flow cytometric analysis of cellular proliferation by CFSE incorporation in FMH9, KL, and LL cells after *Myb* siRNA transfection relative to the corresponding negative control. Continuous lines indicate cells transfected with the negative control siRNA, whereas dashed lines indicate si*Myb*-transfected cells. The statistical analysis was performed using t test on the geometric means of fluorescence intensity for each time point comparing siNEG versus si*Myb*-treated cells, as indicated by the colour-matched bar on top of every peak (**P < 0.01 and *P < 0.05). Each histogram is representative of six independent experiments. **(F)** Flow cytometric analysis of the cell cycle in FMH9, KL, and LL cells was performed by staining with 7-AAD at 48 h after *Myb* siRNA transfection and compared with the negative control. Percentages of cells in G0/G1 are indicated in each histogram. Each plot is indicative of six independent experiments. **(G)** Representative bar plot showing apoptosis analysis performed by Annexin V staining 72 h post *Myb* siRNA transfection in FMH9, KL, and LL cells (***P < 0.001 and **P < 0.01). Each bar plot represents an average of six independent experiments.

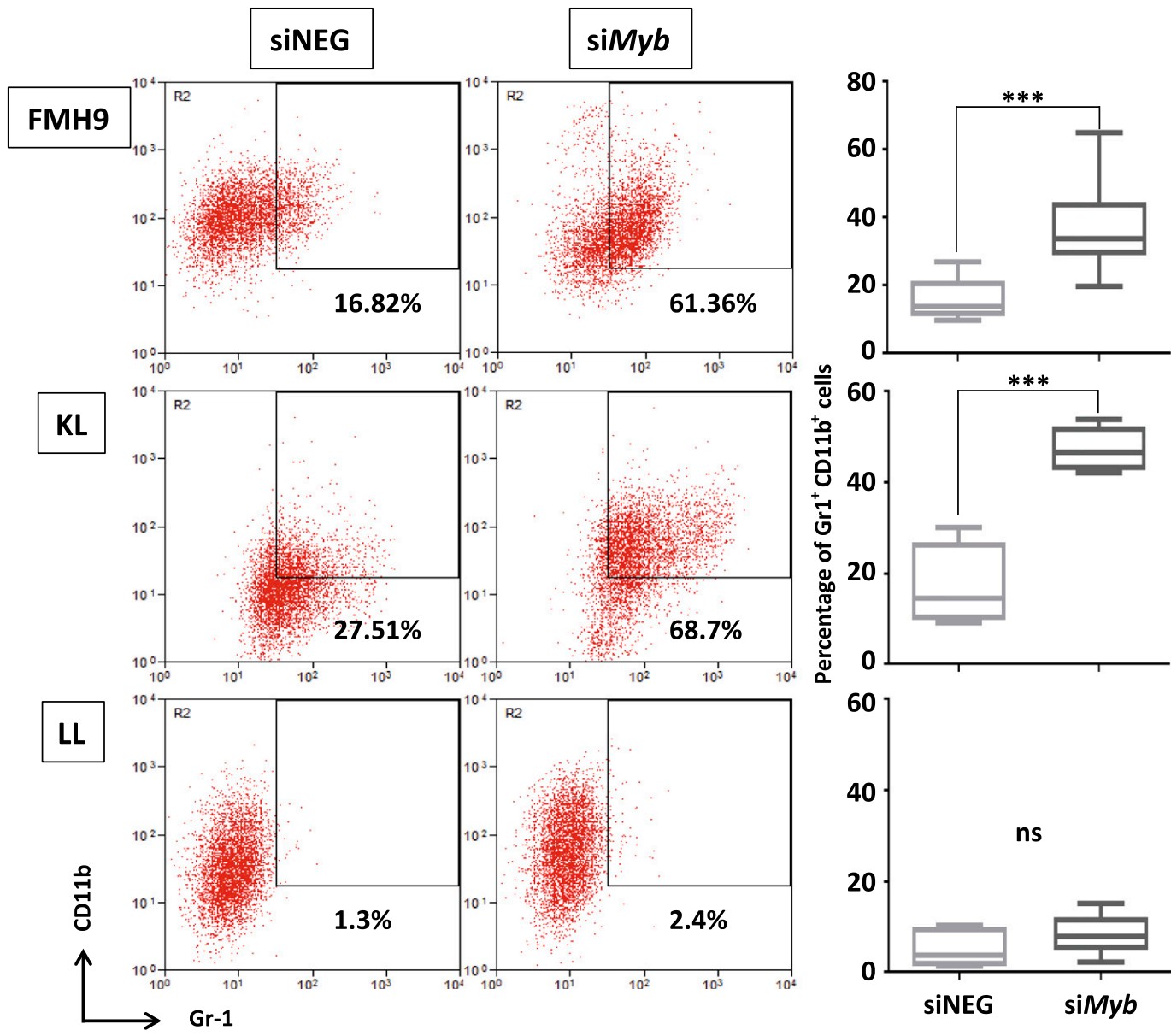

**Figure 2.  Suppression of Myb expression overrides myeloid differentiation block in FMH9, KL but not in LL cells.**
Two-dimensional flow cytometric dot plot representing the analysis of CD11b and Gr-1 myeloid surface markers expression in FMH9, KL, and LL transfected with either *Myb* siRNA or the corresponding control. The percentage of double-positive cells is indicated in every plot. The box plots in the right panel shows an average of six independent experiments. Statistical significance was calculated using *t* test (****P* < 0.001).

leukaemia cells (Lorenzo et al, 2011; Zhao et al, 2011). We wished to explore how the interplay between Myb and wild-type or mutant C/EBPα influences the transcriptome, so we performed RNA-seq analysis following *Myb* knockdown in the context of wild-type or mutant C/EBPα. In line with previous reports, inspection of the RNA-seq datasets for FMH9, KL, and LL cells showed the expected patterns for typical myeloid genes that are known to be Myb targets, such as *Gfi1, Itgam/CD11b, Gr-1/Ly6d,* and *Ccnd2* (Fig S4A and B).

Analysis of the RNA-seq data by global correlation clustering of fragments per kilobase of transcript per million mapped reads (FPKM) in steady-state conditions (i.e., after control siRNA treatment only) revealed higher similarity between KL and LL cells, with FMH9

cells clustering on their own, in line with previous reports on biallelic CEBPA mutants (Wouters et al, 2009; Taskesen et al, 2011) (Fig 3A). Differential gene expression analysis comparing control and si*Myb*-transfected FMH9, KL, and LL cells revealed 790, 1217, and 40 genes being down-regulated, whereas 1364, 1668, and 329 genes were up-regulated (Fig S5A). This demonstrates that, whereas FMH9 and KL cells display a large number of genes responsive to *Myb* knockdown, LL cells exhibited only minor transcriptomic responses. Intersection of the gene expression changes showed generally a greater overlap of up-regulated genes than down-regulated genes (Fig 3B), whereas hierarchical clustering revealed a higher degree of similarity between gene expression changes in FMH9 and KL cells

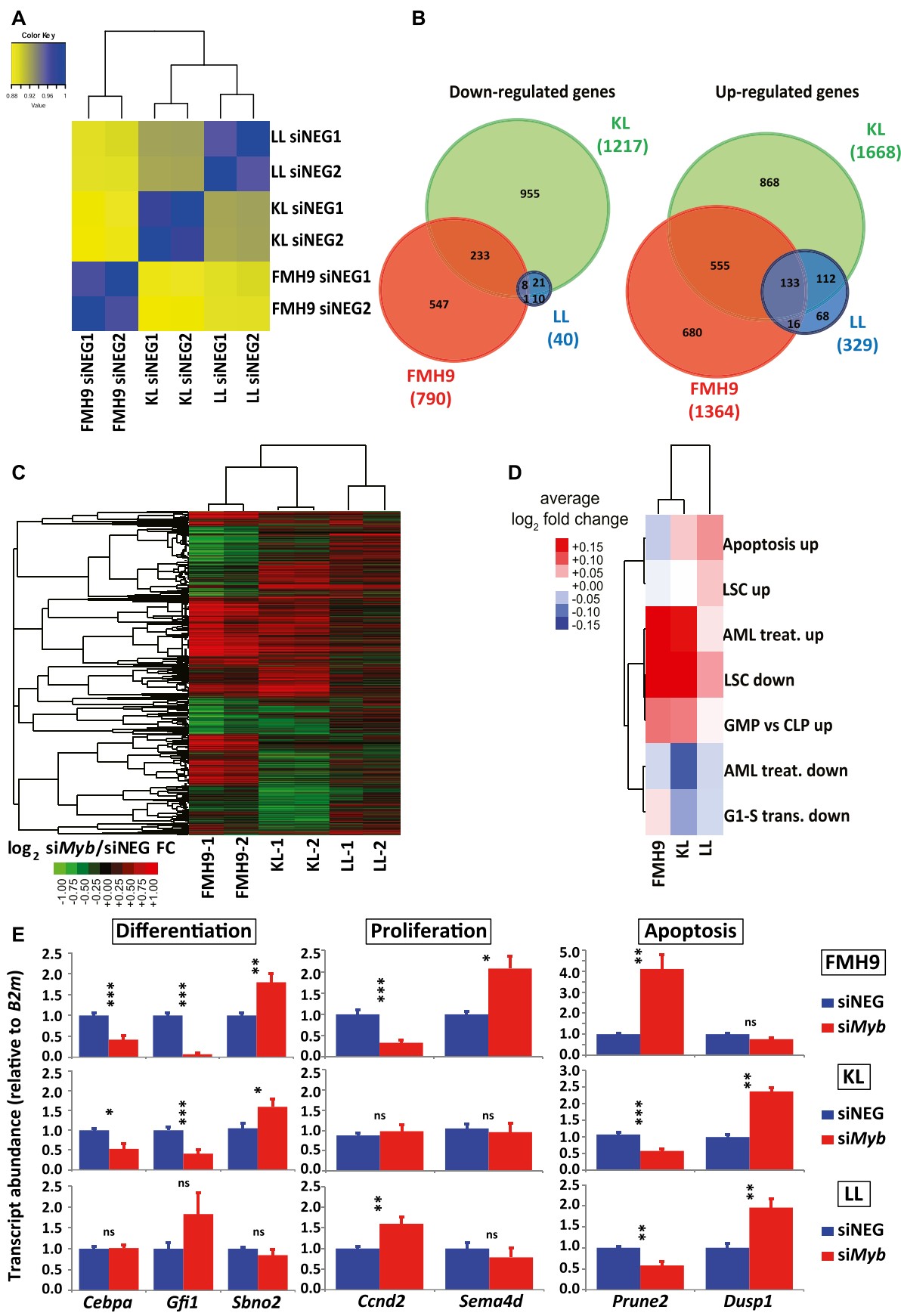

over changes observed in LL cells (Fig 3C). These findings confirm our hypothesis that the LL-mutant leukaemia phenotype is less dependent on Myb.

We next investigated whether *Myb* knockdown results in differential regulation of relevant gene ontologies. We examined negative regulation of G1 to S phase transition (Ashburner et al, 2000), leukaemia stem cell state (Gal et al, 2006), response to AML treatment (Bogni et al, 2006), and terminal myeloid differentiation, computing average log₂ si*Myb*/control fold changes for each ontologies (Table S1). We found that FMH9 and KL cells, but not LL cells, exhibited down-regulation of the leukaemia stem cell gene expression programme following *Myb* knockdown (Figs 3D and S5B). Crucially, this analysis also showed that specifically for FMH9 and KL cells, si*Myb* treatment resulted in up-regulation of genes that are also up-regulated during the treatment of AML.

To confirm the observations obtained from the RNA-seq, we performed quantitative RT-PCR analysis of selected key genes. Significantly, *Cebpa* mRNA was down-regulated upon *Myb* knockdown in both FMH9 and KL cells, but not in LL cells. This suggests that the functional cross-regulation between Myb and C/EBPα could be lost in the presence of biallelic N-terminal *CEBPA* mutations. This analysis confirmed specific regulation by Myb of differentiation-related genes in FMH9 and KL cells but not in LL cells (*Gfi1* and *Sbno2*), changes in the expression of genes leading to a negative impact on proliferation in FMH9 cells (*Ccnd2* and *Sema4d*), and down- and up-regulation of genes related to increased apoptosis seen in KL and LL cells (*Prune2* and *Dusp1*) (Figs 3E and S4B). Moreover, inspection of known Myb target genes revealed a significant repression of *Bcl2* (Taylor et al, 1996; Salomoni et al, 1997), a known anti-apoptotic regulator, in KL cells; concomitantly, we observed the up-regulation of a pro-apoptotic gene normally anti-correlated with *Myb* in AML, namely, *Bcl2l11* (*Bim*) (Jing et al, 2015), in LL cells. This is in agreement with the strong induction of apoptosis observed in these cells upon *Myb* knockdown (Fig S3A and B).

Overall, the transcriptome changes following *Myb* knockdown in cell lines with different CEBPA mutational status are consistent with the corresponding phenotypic changes and further demonstrate that cells with the biallelic LL C/EBPα configuration lack a major dependence on Myb.

## Coincident binding of C/EBPα p42 and Myb correlates with gene repression by Myb, whereas genes positively controlled by Myb tend to bind C/EBPα p30

To shed light on the possible interplay between the mutational status of C/EBPα and the Myb-dependent regulation of C/EBPα target genes, we set out to investigate the chromatin binding properties and transcriptional effects of C/EBPα p30, C/EBPα p42, and K313KK-mutant C/EBPα isoforms. The N-terminal–mutant L

allele (Kirstetter et al, 2008) leads to the expression of the p30 isoform only, whereas the C/EBPα K313KK–mutant allele gives rise to a C-terminal mutant that disables the DNA-binding domain, resulting in a block in differentiation (Bereshchenko et al, 2009). Because the binding dynamics of C/EBPα in double-mutant cells would be technically difficult to characterise, we used previously published chromatin immunoprecipitation sequencing (ChIP-Seq) data from a single isoform transfection model (Grebien et al, 2015). This study used HA-tagged *Cebpa* constructs transfected into the FDCP1 cell line (Bernard et al, 1991) and an immortalised IL3-dependent murine myeloid cell line that approximates to the wild-type C/EBPα leukaemia line FMH9 that we used for the *Myb* knockdown studies. This latter study concurrently provided gene expression microarray analysis of mock- and C/EBPα-transfected cells. As expected, there were significant increases in *Cebpa* transcript abundance following overexpression of C/EBPα p30, C/EBPα p42, and C/EBPα K313KK as compared with mock transfection (Fig S6A, compare with Fig S6B in FMH9, KL, and LL cells). Analysis of ChIP-seq data revealed 17,452, 22,873, and 68,432 peaks for C/EBPα p30-, C/EBPα p42-, and C/EBPα K313KK–transfected cells, respectively, which were mostly located in intergenic and intronic regions (Fig S6C and D). However, visual inspection of the C/EBPα K313KK dataset revealed low signal in K313KK peaks, hinting that those are not specific, consistent with the loss of binding due to the K313KK mutation in the DNA-binding domain (Grebien et al, 2015; Bereshchenko et al, 2009). We next characterised specific p30 and p42 peaks by ranking tag counts around merged summits by p30/p42 fold change and identified 3,585 p42-specific, 19,949 shared, and 4,421 p30-specific peaks (Fig 4A). Although we did not use the C/EBPα K313KK dataset as a direct base for comparison because of the deleterious effect of the K allele on DNA binding, and thus low signal to noise ratio, its binding pattern was mostly located in shared sites. Crucially, by retrieving tag counts for Myb ChIP-Seq datasets, we observed that Myb binding largely parallels that of p42 binding, both in FDCP1 and in an MLL-AF9/NrasG12D murine AML cell line (Roe et al, 2015), although there was an overlap with some regions that predominantly bind p30 (Fig 4A). GO analysis of the p42-specific peaks revealed signalling pathways involved in haematopoietic homeostasis and pro-apoptotic genes (Fig 4B). Conversely, p30-specific peaks were enriched in pluripotency genes, consistent with the leukaemia stem cell signature seen amongst the up-regulated genes following *Myb* knockdown in LL cells (Figs 3D and 4B, and S5B).

To characterise the consequences of C/EBPα binding on gene expression, we performed GSEA using microarray datasets from FDCP1 cells expressing p30, p42, and K313KK, ranking by log₂ fold change against mock transfection. We selected gene sets corresponding to the closest genes of the top 1,000 peaks for cognate ChIP-Seq datasets (p30, p42, and K313KK) in accordance with the constraints of GSEA. We observed significant correlations between

---

**Figure 3. *Myb* down-regulation causes concomitant differential regulation of leukaemia gene expression programmes in both FMH9 and KL but not LL cells.**
**(A)** Spearman correlation clustering of steady-state, control scrambled negative siRNA-transfected FMH9, KL, and LL cells. **(B)** Venn diagram overlaps of differentially expressed genes in FMH9, KL, and LL cell lines following *Myb* knockdown. Left and right: significantly down- and up-regulated genes, respectively. **(C)** Hierarchical clustering of log₂ fold changes resulting from si*Myb* treatment in FMH9, KL, and LL cells. **(D)** Average si*Myb*/siNEG log₂ fold changes for leukaemia-relevant GO classes. **(E)** RT-qPCR gene expression analysis of differentiation, apoptosis, and cell cycle genes post control and si*Myb* transfection. Relative expression values are presented as ± SEM. Statistical analysis was performed using *t* test (\*\*\**P* < 0.001, \*\**P* < 0.01, and \**P* < 0.05). Each bar plot represents an average of six independent experiments.

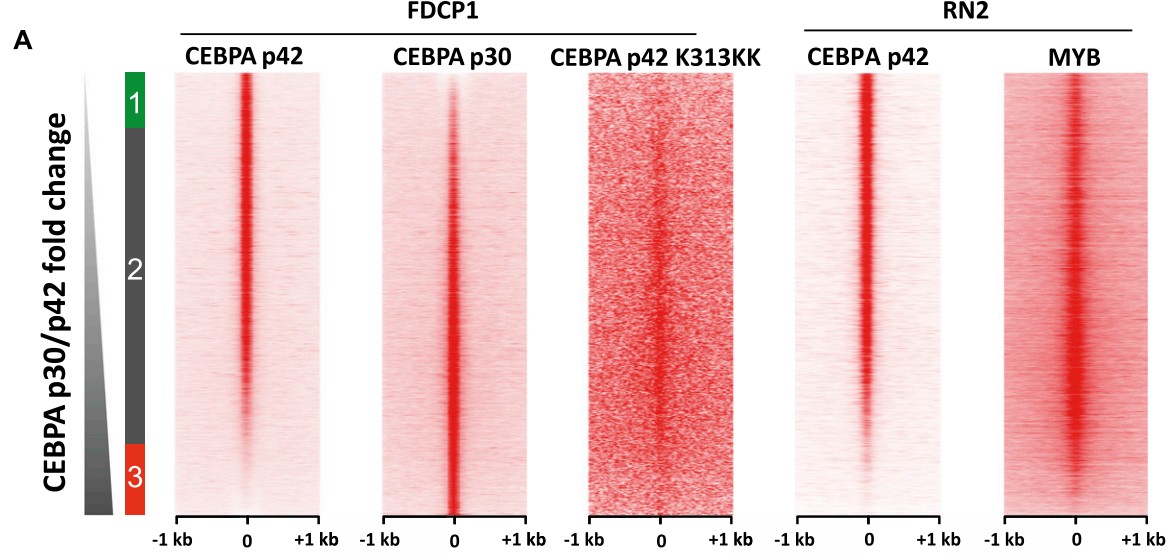

**A**

FDCP1 — CEBPA p42 — CEBPA p30 — CEBPA p42 K313KK

RN2 — CEBPA p42 — MYB

CEBPA p30/p42 fold change — 1 2 3

-1 kb 0 +1 kb (repeated for each panel)

**B**

**Group 1: p42-specific**

-logP
0 · 2.5 · 5 · 7.5 · 10

Wikipathways

B Cell Receptor Signaling Pathway
Adipogenesis
IL–5 Signaling Pathway
TGF Beta Signaling Pathway
Kit Receptor Signaling Pathway
EGFR1 Signaling Pathway
IL–3 Signaling Pathway
IL–2 Signaling Pathway
Apoptosis
Nuclear receptors in lipid metabolism and toxicity

0 · 10 · 20 · 30 · 40
num. genes

**Group 3: p30-specific**

-logP
0 · 4 · 8 · 12 · 16

Podocyte protein–protein interactions
TGF–beta Receptor Signaling Pathway
B Cell Receptor Signaling Pathway
Myometrial Relaxation and Contraction
PluriNetWork
IL–5 Signaling Pathway
Wnt Signaling Pathway NetPath
mRNA processing
IL–3 Signaling Pathway
Leptin Insulin Overlap

■ -logP
□ num. genes

0 · 50 · 100 · 150 · 200
num. genes

**C**

p30 vs mock — Enrichment plot: cebpa_p30_fdcp1_top1000.grp — NES=-1.597 — FWER p=0

K313KK vs mock — Enrichment plot: cebpa_k313kk_fdcp1_top1000.grp — NES=-1.001 — FWER p=0.366

p42 vs mock — Enrichment plot: cebpa_p42_fdcp1_top1000.grp — NES=1.972 — FWER p=0

**D**

FMH9 · KL · LL

CEBPA p30/p42 fold change — 1 2 3

log₂ FC
-0.15 -0.10 -0.05 +0.00 +0.05 +0.010 +0.15

**E**

FMH9 · KL · LL

log₂ FPKM FC
0.3 · 0.2 · 0.1 · 0.0 · −0.1 · −0.2 · −0.3

Groups 1 2 3 · 1 2 3 · 1 2 3

p42-induced gene activation and p42 binding, as well as between p30-induced gene repression and p30 binding (Fig 4C). However, C/EBPα K313KK binding was not correlated with changes in gene expression.

We next asked how C/EBPα isoform binding correlates with genes whose expression is altered by *Myb* knockdown. We plotted gene expression fold changes caused by *Myb* knockdown in FMH9, KL, and LL cells against the ChIP-seq data ranking of p30/p42 binding. Two broad conclusions arise from this analysis: first, Myb-repressed genes largely bound p42 in the presence or absence of p30; second, genes that are positively regulated by Myb are more predominant amongst the group of genes that preferentially bind p30 (Fig 4D and E). Conversely, p30 binding was linked with down-regulation of gene expression upon *Myb* knockdown.

We next investigated the sequence content of p30- and p42-specific sites by performing motif discovery analysis in the sequences corresponding to these peaks. This revealed that C/EBP, AP-1, Ets, Myb, and Runx motifs were highly enriched in p42-specific peaks (Fig S7A). However, the C/EBP motif was not enriched in the p30-specific peaks. Instead, these peaks were enriched in CTCF, Nrf, and Ets motifs. To confirm these trends, we plotted motif matches amongst increasing p30/p42 fold change as above. Strikingly, C/EBP motifs seemed to be restricted to p42-specific and shared sites (Fig S7B). Myb, Runx, AP-1, and Ets motifs also followed this trend. Conversely, CTCF motifs were highly enriched in p30-specific sites and to some extent in shared sites corresponding to lower p42 binding. Nrf, Sp1, and CREB motifs also seemed to follow this trend, but not Elf motifs. To verify binding of cognate factors to these motifs as well as active and inactive transcriptional hallmarks, we made use of publicly available AML ChIP-Seq datasets (Yue et al, 2014; Roe et al, 2015). Our analysis revealed that C/EBPβ, an essential transcription factor in normal myeloid development (Goode et al, 2016), which is able to bind C/EBP motifs as well, also co-localised with p42 (Fig S7C). Furthermore, by retrieving tag counts for the transcription activation hallmark p300, we could also show that p42-specific and shared sites, but not p30-specific sites, correspond to putative enhancer sites, consistent with our GSEA analyses for this isoform (Figs 4C and S7D).

## Discussion

In the present study, we have investigated the requirement for the transcription factor Myb in the maintenance of CN-AML driven by different combinations of *CEBPA* mutations in comparison with leukaemia characterised by the expression of wild-type C/EBPα. We show for the first time that the dependency on Myb is affected by the mutational status of C/EBPα. Compared with cells expressing wild-type C/EBPα, which show a proliferation and differentiation

response to enforced reduction in Myb levels, leukaemia driven by biallelic *CEBPA* mutations exhibits distinct phenotypic responses that are reflected in changes in gene expression. Furthermore, leukaemia with biallelic $N^{ter}/N^{ter}$ *CEBPA* mutations shows a reduced dependency on Myb, whereas $C^{ter}/N^{ter}$ mutant–driven AML cells are as reliant on Myb as those expressing wild-type C/EBPα but exhibit a quite distinct pattern of phenotype and gene expression changes upon *Myb* knockdown.

Here, we show that knockdown of Myb in leukaemia cells harbouring either wild-type or $C^{ter}/N^{ter}$–mutant C/EBPα reverses the abnormal myeloid phenotype normally observed in leukaemia, whereas AML cells carrying biallelic $N^{ter}/N^{ter}$ mutations exhibit persistence of the undifferentiated phenotype. These behaviours are reflected in distinct changes in gene expression. Hence, *Myb* knockdown in leukaemia cells with wild-type or $C^{ter}/N^{ter}$–mutant C/EBPα resulted in the loss of a leukaemia stem cell signature and the up-regulation of a gene expression pattern generally observed in patients that are responding to therapeutic treatment. In line with previous reports (Zhao et al, 2014), our analysis demonstrated *Gfi1* and *Sbno2*, both of which encode transcription factors that can be related to the loss of differentiation block, to be positive and negative targets of Myb activity, respectively. The reduced dependency of $N^{ter}/N^{ter}$–mutant C/EBPα–driven AML cells on Myb is also paralleled by very little change in the number of genes affected by *Myb* knockdown, including no effect on *Gfi1* or *Sbno2*. Interestingly, Myb targets the expression of the *Cebpa* gene itself in the wild-type and $C^{ter}/N^{ter}$ C/EBPα contexts, suggesting positive feedback that is not seen in C/EBPα $N^{ter}/N^{ter}$ cells. This result might provide a hint why LL leukaemia displays a different response to *Myb* manipulation. Analysis of cell cycle and apoptosis also revealed how knockdown of Myb can lead to quite a distinct phenotype; for instance, although no induction in apoptosis/necrosis was observed in C/EBPα wild-type cells, both mutant cell lines displayed a large increase in the percentage of Annexin V+ cells and the appearance of a sub2n population, which is indicative of cells undergoing cell death. However, it is possible that other biological pathways could be affected that lead to such a different response, with the cells perhaps being forced to engage in non-apoptotic fates, such as necroptosis, autophagic cell death, or pyroptosis (Tait et al, 2014); these possibilities remain to be further elucidated.

Considering the different involvement of p30 and p42 in the three leukaemia scenarios we have investigated, the fact that there are p42-only– and p30-only–bound genes indicates that a distinct response should be expected when Myb is reduced, especially when only p30 is present. Most p42- or p42+p30–binding genes also bind Myb, but interestingly, a significant number of p30-only target genes probably do not bind Myb. The genes that bound predominantly p42 fall into GO groups, including those associated with

---

**Figure 4.   p42 C/EBPα binding is linked with gene activation increased following Myb knockdown, whereas p30 C/EBPα binding correlates with gene repression independently of Myb.**
**(A)** Heat maps sorted by C/EBPα p30/p42 tag count fold change of ChIP-seq signals for C/EBPα p42, C/EBPα p42 K313KK, and C/EBPα p30 isoforms in the FDCP1 cell line, as well as for Myb and C/EBPα p42 in the RN2 cell line. **(B)** GO analyses of p42 and p30 C/EBPα-specific peaks (left, right). **(C)** Gene set enrichment analyses of C/EBPα p30, C/EBPα p42, and C/EBPα p42-K313KK binding versus cognate-induced fold change (top left, top right, and bottom left, respectively). **(A, D)** Heat map showing si*Myb*/siNEG gene expression fold change in FMH9, KL, and LL cells sorted by C/EBPα p30/p42 ChIP-seq tag count fold change as in (A). **(A, D, E)** Box plots showing quantification of gene expression fold changes from (D) for the nearest genes from groups 1, 2, and 3 defined in (A). Means indicated.

myeloid homeostasis and differentiation, consistent with the finding that *Myb* knockdown restores myeloid differentiation. Conversely, but in agreement with a persisting leukaemia phenotype not being affected by Myb manipulation, genes bound by p30 include pluripotency genes. This is in line with a previous report that mature cells can tap into stem cell regulatory networks when experiencing mutational hits in key differentiation factors (Soucie et al, 2016).

The vast majority of genes affected by *Myb* knockdown in the context of $N^{ter}/N^{ter}$ biallelic–mutant C/EBPα are de-repressed and overlap with genes similarly affected in the context of $C^{ter}/N^{ter}$. That genes such as *Dusp1* are not responsive to Myb changes in the context of wild-type C/EBPα presumably means that p42, being fully competent to dimerise and bind DNA is less dependent on Myb. Mapping these gene expression changes onto the profile of p42/p30 binding revealed that they can be expected normally to bind both p42 and p30. The large number of genes affected by *Myb* knockdown that distinguished the response of $C^{ter}/N^{ter}$ from $N^{ter}/N^{ter}$ are both down-regulated (1,188 genes) and up-regulated (1,423 genes), the majority of the former corresponding to genes that are preferentially bound by p30. In contrast, the up-regulated genes, normally repressed by Myb as seen in the $N^{ter}/N^{ter}$ situation, tend to be genes that exhibit greater binding by p42 when it is present. Intriguingly, in the $N^{ter}/N^{ter}$ situation, in which most of the genes affected by *Myb* knockdown reflect loss of Myb-dependent repression, these genes also fall into the category of preferential binding to p42. Because no p42 is present in the $N^{ter}/N^{ter}$ leukaemia, this must mean that such genes can be regulated by Myb without a prerequisite for cooperation with C/EBPα p42 or alternatively that another C/EBP family protein such as C/EBPβ can act in place of C/EBPα. In this circumstance, the profile of C/EBPβ binding to genes in the context of myeloid leukaemia cells parallels that of C/EBPα p42. The consequence of this is that most of the genes bound by C/EBPα p42 that are Myb dependent would not exhibit such redundancy of C/EBP protein requirement.

By analysing the occurrence of transcription factor consensus binding motifs in the sequence content of the C/EBPα ChIP-seq peaks, we observed that the regulatory network maintaining the different leukaemia statuses involves different possible sets of transcription factor binding. p42-specific peaks are enriched in C/EBP, Ap1, Myb, Ets, and Runx motifs, whereas the p30-specific peaks contain mostly CTCF, Nrf, and Ets motifs. Importantly, we also observed that p42-specific peaks that are also bound by Myb are highly enriched in the transcription activation hallmark p300. This is to be expected as p300 has been demonstrated to be one of the most important cofactors for Myb and the inhibition of the Myb/p300 interaction is crucial for the maintenance of the leukaemia state (Pattabiraman et al, 2014). The same analysis performed for the p30-specific peaks revealed those sites may be enriched for CTCF in cells that do not express CEBPα p42 (Fig S7D).

In conclusion, we have shown that the nature of mutations in one transcription factor that drives leukaemia can dictate how another leukaemia-associated oncoprotein affects the maintenance of the leukaemia phenotype. The precise nature of the interactions between C/EBPα or its mutated variants and the Myb protein on specific genes that dictate the leukaemia phenotype remain to be elucidated, and it would be fruitful to assess in more detail the relevance of this interaction in human leukaemia patients harbouring those mutations. Added complexity in the involvement of C/EBP proteins in determining the Myb dependency of a given leukaemia most likely goes beyond solely the balance of the C/EBPα isoforms, especially given the possibility that co-expressed factors such as C/EBPβ might compete for binding to C/EBP motifs and heterodimerise with C/EBPα. Ultimately, our findings call for a larger study to determine how manipulating MYB would impact on the maintenance of both murine and human leukaemia driven by other genetic lesions.

## Supplementary Information

## Acknowledgements

The work was supported by a Bloodwise Programme Grant (12010) held by J Frampton, S Dumon, and P Garcia and through funding provided by the College of Medical and Dental Sciences of the University of Birmingham. This project has received funding from the European Research Council under the European Union's Horizon 2020 research and innovation programme (grant agreement no. 636855/StG to F Grebien)

### Authors Contributions

G Volpe: conceptualisation, data curation, investigation, and writing—original draft.
P Cauchy: data curation, formal analysis, and writing—original draft.
DS Walton: investigation.
C Ward: formal analysis.
D Blakemore: investigation.
R Bayley: investigation.
ML Clarke: investigation.
L Schmidt: investigation.
C Nerlov: resources.
P Garcia: funding acquisition and writing—review and editing.
S Dumon: conceptualisation, supervision, and funding acquisition.
F Grebien: conceptualisation, formal analysis, investigation, and writing—review and editing.
J Frampton: conceptualisation, supervision, funding acquisition, project administration, and writing—review and editing.

### Conflict of Interest Statement

The authors declare no conflicts of interest.

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
