## [Reviewer comments · Life Science Alliance]

Life Science Alliance

Dependence on Myb expression is attenuated in myeloid leukaemia with N-terminal CEBPA mutations

Jon Frampton, Giacomo Volpe, Pierre Cauchy, David Walton, Carl Ward, Daniel Blakemore, Rachael Bayley, Mary Clarke, Luisa Schmidt, Claus Nerlov, Paloma Garcia, Stéphanie Dumon, and Florian Grebien

DOI: <https://doi.org/10.26508/lsa.201800207>

Corresponding author(s): Jon Frampton, Institute of Biomedical Research

Review Timeline:

Submission Date:	2018-10-07
Editorial Decision:	2018-11-06
Revision Received:	2018-12-19
Editorial Decision:	2019-01-28
Revision Received:	2019-02-15
Editorial Decision:	2019-02-28
Revision Received:	2019-03-07
Accepted:	2019-03-07

Scientific Editor: Andrea Leibfried

Transaction Report:

November 6, 2018

Re: Life Science Alliance manuscript #LSA-2018-00207-T

Prof. Jon Frampton
Institute of Biomedical Research
College of Medical and Dental Sciences
University of Birmingham
Edgbaston
Birmingham B15 2TT
United Kingdom

Dear Dr. Frampton,

Thank you for submitting your manuscript entitled "Dependence on Myb expression is attenuated in myeloid leukaemia with N-terminal CEBPA mutations" to Life Science Alliance. The manuscript was assessed by expert reviewers, whose comments are appended to this letter.

As you will see, the reviewers appreciate your work. However, they also think that a lot of issues need to get addressed to make your manuscript of value to others. Given this input, we would like to invite you to submit a revised version, following the reviewers suggestions on how to better present your data to allow others to recapitulate and replicate the analyses performed. Reviewer #2 and #3 also make comments on how the work could be extended to further strengthen your work. We think that there is merit in the dataset per se if properly presented, so extending the work is not mandatory for acceptance here.

Thank you for this interesting contribution to Life Science Alliance. We are looking forward to

receiving your revised manuscript.

Sincerely,

- A letter addressing the reviewers' comments point by point.
- An editable version of the final text (.DOC or .DOCX) is needed for copyediting (no PDFs).
- High-resolution figure, supplementary figure and video files uploaded as individual files: See our detailed guidelines for preparing your production-ready images, <http://life-science-alliance.org/authorguide>
- Summary blurb (enter in submission system): A short text summarizing in a single sentence the study (max. 200 characters including spaces). This text is used in conjunction with the titles of papers, hence should be informative and complementary to the title and running title. It should describe the context and significance of the findings for a general readership; it should be written in the present tense and refer to the work in the third person. Author names should not be mentioned.

B. MANUSCRIPT ORGANIZATION AND FORMATTING:

Full guidelines are available on our Instructions for Authors page, <http://life-science-alliance.org/authorguide>

Reviewer #1 (Comments to the Authors (Required)):

Volpe and colleagues have explored the consequences of loss of c-Myb expression to AML cell growth and differentiation across the spectrum of C/EBP α mutations. The authors developed two hematopoietic progenitor cell lines with homozygous N- or heterozygous N/C-terminal C/EBP α mutations. AML with no C/EBP α mutation was represented by the previously described FMH9 cell line. No reference is offered for FMH9 cells. Loss of c-Myb was induced by siRNA mediated knock down (KD). The sum of these experiments suggests that AML is less dependent on c-Myb expression in cells carrying homozygous N-terminal mutations, which is a very interesting finding. However, the observation is limited to a single cell line and, strictly speaking, dependence or lack of dependence of AML on c-Myb has not been demonstrated in these studies as effects on differentiation, survival and proliferation have been limited to in vitro experiments.

The authors subsequently explored how the interplay between c-Myb and C/EBP α influences transcription in the C/EBP α mutant cell lines. Comparison of RNA-seq data demonstrated that the LL phenotype is less dependent on c-Myb than FMH9 and KL cells and that FMH9 and KL cells, but not LL cells, display downregulation of the leukemia stem cell gene expression program following Myb knockdown as well as up regulated the expression of genes that increase expression during treatment of AML. These experiments are consistent with the phenotypic changes associated with different C/EBP α mutations demonstrated by the authors. The authors then examine the relationship between C/EBP α mutational status and c-Myb regulation of C/EBP α regulated genes. These experiments mine publically available data bases from a study that used HA-tagged C/EBP α mutants transfected into cell lines and determined that p42 binding largely paralleled c-Myb binding, less so with p30. The major points from this analysis are that Myb-binding largely paralleled that of p42, GSEA allowed correlation between p42-induced genes and p42 binding and roughly the same for p30 induced gene expression and p30-binding. Further, analysis gene expression in FMH9, KL and LL cells after c-Myb KD demonstrated that c-Myb repressed genes mainly associated with p42 binding while genes positively regulated by c-Myb were among those associated with p30-binding.

In summary, the authors provide correlative association between gene expression and C/EBP α mutation status in the context of c-Myb binding and as related to phenotypic information gained from FMH9, KL and LL cells. While it is unfortunate that the authors did not make an attempt to extend their findings to human AML and are limited to observations in three cell lines, each representing a subset of C/EBP mutations in AML, these studies will be of interest to a relatively broad group of investigators that study cancer, differentiation and gene regulation, particularly among groups who study c-Myb and its relationship to cancer and differentiation.

Several issue that need be addressed:

1. The Methods section is inadequate and needs considerable work. For example, the authors appear to have carried out RNA-seq experiments with material from their FMH9, KL and LL cells but there is section detailing RNA-seq in the Methods and no description of how material was prepared for RNA-seq or where the RNA-seq was carried out (in house? Core facility?, outside vendor?). If the RNA-seq data came from a previous manuscript published by this group it needs to be clearly stated as such and cited. If the RNA-seq data is new, it needs to be submitted to a public repository with clear indication of accession number. If the RNA-seq data has been previously published a reference needs to be provided as well as the data respository where the data resides and the accession number.

2. There is essentially no information describing how the bioinformatics analysis was carried out despite a mention that it would be detailed in the Supplemental Data. I was not able to find this information in the Supplemental data. Mention of bioinformatics packages in the Results section is

not adequate.

3. Lists of gene signatures from GSEA analysis in the Supplementary data are not legible. These lists need to be included as tables in the supplementary data section (Figure S2B).
4. Much of the work in this paper utilizes publically available data bases. The authors generally provide references but leave it up to the reader or those who would follow up on this work to find out how to access them. The authors need to provide not only references but state where these data bases reside and precise accession numbers. This information should go into the methods section where Bioinformatics is discussed. Further, cells, cell lines or mice that serve as the basis for the data mined from public data bases need to be clearly described in the Methods section and clearly associated with the data bases as well as appropriate references.
5. The manuscript utilizes cell lines that were made by the authors, apparently for this manuscript, by serial replating of E14.5 fetal liver cells carrying knock-in modifications. However, there is no discussion of the protocol in the Methods section or the origin of the mice that served as the source of fetal liver cells. The mice (reference, strain, genotype, source), as well as the protocol used to generate the cell lines, need to be described in the Methods section. Also, it is appropriate to provide cell densities at the time of harvest for RNA-seq as well as other experiments.
6. The source and fluorochrome tags on the antibodies used for flow cytometry are provided but not the clone designations. The type and model of flow cytometer as well as software used for analysis should also be provided in the Methods.
7. No mention is made of how dye dilution experiments were carried out in the Methods section, not even the dye that was used. In addition, the analysis of these experiments is minimal, simply presentation of curves. It would appear that there are shifts in mean fluorescence intensity of the curves between siControl and siMyb presented in Figure 1E though it is not clear if these are significant. In particular, the siMyb 72 hr population appears bimodal but there is no mention or discussion of this point. There are standard approaches to analyze dye dilution data.
8. It is unclear in Figure 1A what samples are actually different as indicated by ANOVA. These should be specifically marked on the figure and the reference to ANOVA can be moved to the figure legend. The graph in Figure 1A is not a box graph as it is described in the body of the manuscript.
9. Figure 1B provides information about c-Myb mRNA expression in the FMH9, KL and LL cell lines but it is not clear if the differences between cell lines are statistically significant.
10. No statistics are provided in Figure 1D.
11. Annexin V staining presented in Figure 1E was used to assess potential apoptosis. However, it is not clear if the Annexin V+ PI- (or 7AAD- depending on which intercalating dye was used), Annexin V+ PI+ or total Annexin V binding is being reported. These populations provide different pieces of information. Also, it would be appropriate to provide a second measure of apoptosis such as Caspase activation, which can readily be done by flow cytometry using an anti-active caspase antibody.
12. DNA content is measured by staining cells with 7AAD and bar graphs providing the percent cells in G0/G1 are presented in Figure 1F. Representative flow cytometry for each cell line should be provided in the Supplementary Data so the reader can judge the quality of the data. The authors

should also indicate if they gated on live cells or displayed the entire sample. No information is provided regarding S- or G2/M-phase.

Reviewer #2 (Comments to the Authors (Required)):

One group of human AMLs carries and is driven by mutations in CEBPA. There are two main types of mutations - those affecting the N- and C-terminal regions of the CEBPA transcription factor, respectively, and these are most commonly found together, ie in separate alleles. It is also known that AMLs of most/all subtypes are dependent on MYB and sensitive to suppression of MYB activity. The main finding in this paper is that, as indicted by the title, murine AML cell lines with either bi-allelic N-terminal CEBPA mutations ("LL"), or carrying one mutant allele of each type ("KL") , show reduced sensitivity to siRNA-mediated Myb knockdown. These data re complemented by gene expression studies to identify genes affected by Myb knockdown in each case compared to cells with WT Cebpa. The second part of the paper is an analysis of published CHIP-Seq data that focuses on genes bound by the two CEBPA isoforms, p30 and p42, that are affected by N and C-terminal mutations respectively.

1. While the main conclusion is the most interesting part of the m/s, I have a number of concerns regarding the magnitude and significance of the effect seen. The "resistance" to MYB knockdown seen with the KL lines that represent the most common type of CEBPA-mutant AML is much less marked than that seen with the LL line that represents bi-allelic N-terminal mutant AML, which is rarely if ever seen in human disease. Indeed Fig 2 shows no resistance to Myb knockdown-induced differentiation in the case of the KL line. The magnitude of the proliferative effect is also hard to gauge since there is only a small increase in G0/G1 cells even in the control line (numbers are not given but it looks like ~ 58% to ~ 64%). THE CFSE data are a bit more convincing but even there the effect in the control cells is quite small at the time point examined. It may be useful to look at later time points.

Moreover there is a large increase in apoptosis seen in the LL and KL cells on Myb knockdown which isn't seen in the control. There is no examination or discussion of why this might be - this should be included. For example, is Bcl2, a known MYB target, down-regulated in these cell lines upon Myb knockdown?

2. The gene expression studies in this system show a very different pattern for the LL cells of the KL cells which, as the authors say does suggest a difference in MYB dependency of many genes. How this is can be interpreted though is hard to know, because we have no idea about the relative chromatin occupancy of sites by the 2 mutant alleles. Indeed this is something of a problem for the whole paper, because it relies on speculative inferences based on chromatin occupancy by the WT isoforms (from the second part of the paper).

3. It is not clear how the KL and LL lines were derived - was it from published knock-in mice? Much more detail is needed.

4. It is unclear how good a control FMH9 is for the KL and LL lines. How comparable are they with respect to lineage stage and potential? The data of Fig S1B suggest they may not be that well matched because of the large differences in gene expression in the absence of Myb knockdown. Similarly, the CHIP data on P42 and p30 occupancy are derived from yet another cell line - FDCP1.

5. There are a lot of other important details missing. For example, what are the data sets used in

the second part of the m/s from? References to papers are provided, but GEO or similar accession numbers for the data actually used are essential. Also it is unclear where the "LSC Programme" dataset (Fig S2B) is from.

6. A couple of more minor points: (i) The Discussion on p8 top paragraph cites a Ref 54, but the last reference in the reference list is # 42. (ii) GFi1 is actually a MYB target gene (Zhao et al Oncogene, 33:4442-44492 (2014)), it doesn't just "appear" to be positively regulated by MYB (p7).

Overall, the most positive aspect of the paper is that it provides the first clear example of an AML type with reduced Myb dependency at least with regard to proliferation, and differentiation in the case of the LL line. Has some implications for human AMLs, since even KL cells showed some Myb independent proliferation. It would be good to see if this observation is replicated in human AML.

The second part of the paper though is somewhat hard to follow. A major difficulty is that the first part of the paper deals with the effects of mutant CEBPA while the latter deals with WT. The analyses and interpretation of the previously-published data on p30 and p42 chromatin occupancy would benefit from consideration of the functional properties of p30 cf p42 proteins and mutants thereof in interpreting the results.

Reviewer #3 (Comments to the Authors (Required)):

This study examines the effect of Myb knockdown in the context of different Cebpa mutations. Differential effects were seen, and Myb knockdown failed to override the block in myeloid differentiation in cells with biallelic N-terminal mutations. Evidence from gene expression and CHIP-seq datasets suggests a functional co-operation between C/EBP α and Myb that breaks down in the presence of two N-terminal mutations. This is a well written study that provides useful new information.

Comments

1. Figure 1A. MYB expression levels are highest in cases carrying biallelic mutations. It is unclear how cases with multiple mutations have been included in this analysis. Since the data do not appear to be normally distributed the analysis should be performed using a non-parametric test.
2. Presumably the cells lines employed in this study were clonal? Why were experiments not performed using more than one clone for each mutational state to verify the findings? And why were the effects of single N-terminal or C-terminal mutations not investigated?

POINT BY POINT ANSWERS TO THE REVIEWERS' COMMENTS

Reviewer #1 (Comments to the Authors (Required)):

1. The Methods section is inadequate and needs considerable work. For example, the authors appear to have carried out RNA-seq experiments with material from their FMH9, KL and LL cells but there is section detailing RNA-seq in the Methods and no description of how material was prepared for RNA-seq or where the RNA-seq was carried out (in house? Core facility?, outside vendor?). If the RNA-seq data came from a previous manuscript published by this group it needs to be clearly stated as such and cited. If the RNA-seq data is new, it needs to be submitted to a public repository with clear indication of accession number. If the RNA-seq data has been previously published a reference needs to be provided as well as the data repository where the data resides and the accession number.

We thank the reviewer for pointing this out as the Supplementary Information file was accidentally deleted. All the information regarding the RNA-seq libraries preparation and genomic sequence facility used has been included in the material and methods section and in the supplementary information section under the "RNA-sequencing" subheading. The details regarding the availability of the data generated in this study and the respective GEO accession number has been included in the material and methods section under "Data availability". Information regarding the previously published RN2 C/EBP α data as well as other published ChIP-Seq data used in the manuscript has been included under "ChIP-Seq data processing" and "Published ChIP-Seq data processing" in the supplementary material and related references have been added.

2. There is essentially no information describing how the bioinformatics analysis was carried out despite a mention that it would be detailed in the Supplemental Data. I was not able to find this information in the Supplemental data. Mention of bioinformatics packages in the Results section is not adequate

As requested by the reviewer, we have included a detailed explanation about the bioinformatic analysis for both RNA-seq and ChIP-seq in the Supplementary Information.

3. Lists of gene signatures from GSEA analysis in the Supplementary data are not legible. These lists need to be included as tables in the supplementary data section (Figure S2B)

Following the reviewer's comment, we have included the list of genes from the GSEA analysis in Supplementary Table S2).

4. Much of the work in this paper utilizes publicly available data bases. The authors generally provide references but leave it up to the reader or those who would follow up on this work to find out how to access them. The authors need to provide not only references but state where these data bases reside and precise accession numbers. This information should go into the methods section where Bioinformatics is discussed.

As requested by the reviewer, we have included a more detailed explanation of the data processing, with GEO accession number and related references in the Supplementary

Information under “Published ChIP-Seq data processing” and “Published gene expression analysis” subheadings. ChIP data from the occupancy of p30 and p42 in the FDCP1 line can be obtained from the Grebien Lab upon request.

Further, cells, cell lines or mice that serve as the basis for the data mined from public data bases need to be clearly described in the Methods section and clearly associated with the data bases as well as appropriate references.

We agree with the reviewer on the need to clarify the information regarding the use of primary cells, cell lines or mice on the different data sets publicly available that were used on this article. Accordingly, all the necessary information has been included in the methods section under “Cell lines” sub-heading together with the appropriate references.

5. The manuscript utilizes cell lines that were made by the authors, apparently for this manuscript, by serial replating of E14.5 fetal liver cells carrying knock-in modifications. However, there is no discussion of the protocol in the Methods section or the origin of the mice that served as the source of fetal liver cells. The mice (reference, strain, genotype, source), as well as the protocol used to generate the cell lines, need to be described in the Methods section.

In response to the reviewer’s comments we have (i) added a brief explanation about the origin of the cells and establishment of the cell line, (ii) included the references of the original mouse models from which the cells were derived, and (iii) included a reference to the study in which the cells lines have been generated. All this information has been included under the sub-heading “Cell lines” of the Material and Methods section.

Also, it is appropriate to provide cell densities at the time of harvest for RNA-seq as well as other experiments.

Following the reviewer’s advice, we have added the information regarding the cell densities in the “Transfection experiments, cell viability, proliferation, apoptosis and differentiation assay” sub-heading of the Materials and Methods section.

6. The source and fluorochrome tags on the antibodies used for flow cytometry are provided but not the clone designations. The type and model of flow cytometer as well as software used for analysis should also be provided in the Methods.

We have added the details requested under the “Transfection experiments, cell viability, proliferation, apoptosis and differentiation assay” sub-heading of the Materials and Methods section.

7. No mention is made of how dye dilution experiments were carried out in the Methods section, not even the dye that was used. In addition, the analysis of these experiments is minimal, simply presentation of curves. It would appear that there are shifts in mean fluorescence intensity of the curves between siControl and siMyb presented in Figure 1E though it is not clear if these are significant. In particular, the siMyb 72 hr population

appears bimodal but there is no mention or discussion of this point. There are standard approaches to analyze dye dilution data.

In response to the reviewer's comment, we have performed statistical analysis using two-tailed paired Student's t-test ($p=0.0062$) for this particular experiment and have included this information in the corresponding histogram and figure legend of Figure 1E and in the Materials and Methods. This statistical test reinforces the difference in mean fluorescent intensity between siNeg and siMyb. Regarding the bimodal proliferation pattern displayed by FMH9 in response to Myb knock-down, our explanation is that this effect could be due in part to the cells being induced to differentiate and in part by the transient effect of the siRNA-mediated knock-down. This possible explanation has been added in the Results section in the paragraph "Manipulation of Myb expression does not reverse the differentiation block in cells carrying biallelic N-terminal CEBPA mutations".

8. It is unclear in Figure 1A what samples are actually different as indicated by ANOVA. These should be specifically marked on the figure and the reference to ANOVA can be moved to the figure legend. The graph in Figure 1A is not a box graph as it is described in the body of the manuscript.

We thank the reviewer for this comment. Upon addressing this comment, we realized that the statistical analysis presented in Figure 1A was not appropriate as those samples are not normally distributed; for this purpose, we have now performed statistical analysis using a non-parametric Kruskal-Wallis test ($p=0.0015$). This is now described in the legend of Figure 1 and in the Material and Methods in the section "Statistical analysis". The legend of Figure 1A has been modified to correct box plot with scatter plot.

9. Figure 1B provides information about c-Myb mRNA expression in the FMH9, KL and LL cell lines but it is not clear if the differences between cell lines are statistically significant.

We have incorporated more data and performed a two-tailed paired Student's t-test ($p=0.015$ and $p=0.003$, respectively). The details of statistical analysis are provided in the legend of Figure 1 and in the materials and methods section

10. No statistics are provided in Figure 1D

As requested by the reviewer, the details of the statistical analysis have been added both in panel 1D and in the corresponding figure legend.

11. Annexin V staining presented in Figure 1E was used to assess potential apoptosis. However, it is not clear if the Annexin V+ PI- (or 7AAD- depending on which intercalating dye was used), Annexin V+ PI+ or total Annexin V binding is being reported. These populations provide different pieces of information. Also, it would be appropriate to provide a second measure of apoptosis such as Caspase activation, which can readily be done by flow cytometry using an anti-active caspase antibody.

We apologize that the apoptosis data presented in Figure 1 are not explained clearly. For this study we performed total Annexin V binding. To provide a clearer view of our data, a

representative histogram of the Annexin V staining in the three different cell lines following siMyb knockdown has been added in Supplementary Figure S2A. For this study we performed total Annexin V binding without the use of a DNA intercalating dye, thus we cannot distinguish between necrotic versus apoptotic cells. In support of our Annexin V data, we have included a graph-bar showing the differential regulation of *Bcl2*, an anti-apoptotic gene, and *Bim* (*Bcl2l11*), a pro-apoptotic gene in WT and CEBPA mutant cells transfected with siMyb or corresponding negative control (Supplementary Figure S2B). This data shows that the pro-apoptotic gene Bim, is up regulated upon *Myb* KD in KL and LL cells whilst anti-apoptotic gene Bcl2 is down regulated in response to *Myb* KD in KL cells only, in agreement with the strongest induction of apoptosis observed in those cells upon *Myb* down regulation. This new data has been included in the text under the subheading “Molecular consequences of *Myb* manipulation carrying either wild type or mutant CEBPA” and related references have been inserted. We hope the reviewer would now find these data more convincing.

12. DNA content is measured by staining cells with 7AAD and bar graphs providing the percent cells in G0/G1 are presented in Figure 1F. Representative flow cytometry for each cell line should be provided in the Supplementary Data so the reader can judge the quality of the data. The authors should also indicate if they gated on live cells or displayed the entire sample. No information is provided regarding S- or G2/M-phase.

As requested by the reviewer we have added a representative histogram for the 7-AAD staining of each cell line transfected with siMyb and compared to the corresponding negative control (Supplementary Figure S1B). Furthermore, we have added a bar plot from a BrdU incorporation experiment indicating the change in the percentages of cells in G0/G1-, S- and G2/M-phase in response to *Myb* downregulation in the three different cell lines (Supplementary Figure S1C). Our data show an increase in the percentage of cells in the G0/G1 phase as a result of *Myb* KD, with a concomitant reduction of cells in both S- and G2/M phases in FMH9 cells only. No cell cycle defects have been observed in CEBPA mutant KL and LL cell lines. In those experiment the gating was performed in live cells. This information has been included in the Supplementary Information.

Reviewer #2 (Comments to the Authors (Required)):

1. While the main conclusion is the most interesting part of the m/s, I have a number of concerns regarding the magnitude and significance of the effect seen. The "resistance" to MYB knockdown seen with the KL lines that represent the most common type of CEBPA-mutant AML is much less marked than that seen with the LL line that represents bi-allelic N-terminal mutant AML, which is rarely if ever seen in human disease. Indeed Fig 2 shows no resistance to *Myb* knockdown-induced differentiation in the case of the KL line. The magnitude of the proliferative effect is also hard to gauge since there is only a small increase in G0/G1 cells even in the control line (numbers are not given but it looks like ~ 58% to ~ 64%). THE CFSE data are a bit more convincing but even there the effect in the control cells is quite small at the time point examined. It may be useful to look at later time points. Moreover, there is a large increase in apoptosis seen in the LL and KL cells on *Myb* knockdown which isn't seen in the control. There is no examination or discussion of

why this might be - this should be included. For example, is Bcl2, a known MYB target, down-regulated in these cell lines upon Myb knockdown?

As commented by the reviewer, we have further examined the cell cycle changes between the different cell lines in the presence of reduced Myb levels and provided the percentage of cells in G0/G1 (Supplementary Figure S1B). In the case of the control line, an increase in G0/G1 from 51% to 74% with concomitant decrease in S-phase and G2/M phases was observed (Supplementary Figure S1C). Statistical analysis has also been included to reinforce the cell cycle changes observed in response to *Myb* KD in FMH9 cells (see Supplementary Figure S1C).

Regarding the apoptosis data, we have added in Supplementary Figure S2B a plot showing the differential regulation of *Bcl2*, a known Myb target and apoptosis regulator. This shows that *Bcl2* is down regulated in response to *Myb* KD in KL cells only, in agreement with the strongest induction of apoptosis observed in those cells upon *Myb* down regulation. We have also added a plot showing the expression of *Bcl2l11* (*Bim*), another regulator of apoptosis that has been shown to be anti-correlated with *Myb* in leukaemia. This has been added in the text under the subheading “Molecular consequences of *Myb* manipulation carrying either wild type or mutant CEBPA” and relative references have been inserted. We hope that the reviewer would now find these data more convincing.

2. The gene expression studies in this system show a very different pattern for the LL cells of the KL cells which, as the authors say does suggest a difference in MYB dependency of many genes. How this is can be interpreted though is hard to know, because we have no idea about the relative chromatin occupancy of sites by the 2 mutant alleles. Indeed, this is something of a problem for the whole paper, because it relies on speculative inferences based on chromatin occupancy by the WT isoforms (from the second part of the paper).

We agree with the reviewer that unravelling the binding dynamics of C/EBP α in WT and CEBPA mutant AML cell lines would have improved the manuscript tremendously and we understand the concerns raised by the reviewer as our data is based on chromatin occupancy by the WT isoform. To overcome this limitation, we sought to use previously published data that have been generated using FDCP1 expressing the p30 isoform only (which is essentially mimicking the expression pattern seen in the N-terminal mutant cells – i.e. LL), p42 only and have added data from a mutant version of the p42 isoform (namely K313KK), which disables the DNA binding domain, thus mimicking the pattern observed in C-terminal mutant cells. The expression of those isoforms, either WT or mutant, lead to an increase of C/EBP α expression that is similar to the expression pattern observed when comparing WT and CEBPA mutant AML cell lines used in this study. To provide a clearer explanation of our data, we have now added a comparison between these three isoforms in Figure 4A, 4C, 5A, 5C and 5D.

3. It is not clear how the KL and LL lines were derived - was it from published knock-in mice? Much more detail is needed.

Indeed, those cells lines have been derived from published knock-in mice generated by the group of Prof Claus Nerlov (Kirstetter et al, 2008 *Cancer Cell*; Bereshchenko O et al, 2010, *Cancer Cell*).

In response to the reviewer's comments we have added a brief explanation about the origin of the cells and establishment of the cell line together with the references to the original mouse models from which the cells have been derived and a reference of the study in which the cells lines have been generated. This information has been added in the "Cell lines" section of the Material and Methods.

4. It is unclear how good a control FMH9 is for the KL and LL lines. How comparable are they with respect to lineage stage and potential? The data of Fig S1B suggest they may not be that well matched because of the large differences in gene expression in the absence of Myb knockdown. Similarly, the ChIP data on P42 and p30 occupancy are derived from yet another cell line - FDCP1.

In response to the reviewer's comment we have added a phenotypic characterization of the cell lines used in this study. This information is provided in Supplementary Figure S1A.

5. There are a lot of other important details missing. For example, what are the data sets used in the second part of the m/s from? References to papers are provided, but GEO or similar accession numbers for the data actually used are essential. Also it is unclear where the "LSC Programme" dataset (Fig S2B) is from.

We agree with the reviewer on the importance of providing GEO accession numbers for publicly available data. This has been added in the Supplementary Information in the section "Published ChIP-Seq data processing". ChIP data from the occupancy of p30 and p42 have been obtained upon request from the Grebien Lab. The reference about the LSC program has been added to the text (Ref 37, Gal, H et al. Gene expression profiles of AML derived stem cells; similarity to hematopoietic stem cells. *Leukemia* **20**, 2147-2154, doi:10.1038/sj.leu.2404401 (2006)).

6. A couple of more minor points: (i) The Discussion on p8 top paragraph cites a Ref 54, but the last reference in the reference list is # 42. (ii) GFi1 is actually a MYB target gene (Zhao et al *Oncogene*, 33:4442-44492 (2014)), it doesn't just "appear" to be positively regulated by MYB (p7).

This mistake has been corrected according to the reviewer's comment and the correct reference (Zhao et al, 2014, *Oncogene*) has been added in the Discussion.

Overall, the most positive aspect of the paper is that it provides the first clear example of an AML type with reduced Myb dependency at least with regard to proliferation, and differentiation in the case of the LL line. Has some implications for human AMLs, since even KL cells showed some Myb independent proliferation. It would be good to see if this observation is replicated in human AML.

We agree with the reviewer that the replication of our results in human cells would be of fundamental importance and would boost the value of our manuscript. However, we have

not managed to obtain any access to primary human AML material from patients carrying CEBPA mutations.

The second part of the paper though is somewhat hard to follow. A major difficulty is that the first part of the paper deals with the effects of mutant CEBPA while the latter deals with WT. The analyses and interpretation of the previously-published data on p30 and p42 chromatin occupancy would benefit from consideration of the functional properties of p30 cf p42 proteins and mutants thereof in interpreting the results.

The answer to this comment has already been provided in point 2.

Reviewer #3 (Comments to the Authors (Required)):

1. Figure 1A. MYB expression levels are highest in cases carrying biallelic mutations. It is unclear how cases with multiple mutations have been included in this analysis. Since the data do not appear to be normally distributed the analysis should be performed using a non-parametric test.

We thank the reviewer for this comment. The data presented in Figure 1A show subgroups of patients that have been clustered by a single mutation only. These data are indeed not normally distributed and therefore we have performed a non-parametric Kruskal-Wallis test ($p=0.0015$). This has also been corrected in the figure legend and in the Statistical Analysis section of the Materials and Methods.

2. Presumably the cells lines employed in this study were clonal? Why were experiments not performed using more than one clone for each mutational state to verify the findings? And why were the effects of single N-terminal or C-terminal mutations not investigated?

The description of both KL and LL cell lines has been added in the supplementary information under the subheading "Cell Lines". The cells were generated by the Grebien group by picking single clones after serial replating in semisolid medium followed by continuous liquid culture. We had only one cell line per genotype available at the time the experiments were conducted. Moreover, we chose not to use single N-terminal or C-terminal mutant cell lines as those monoallelic mutations are not leukaemogenic and are very rare events in AML (Kirstetter P. et al, 2008; Bereshchenko O. et al, 2010).

January 28, 2019

Re: Life Science Alliance manuscript #LSA-2018-00207-TR

Prof. Jon Frampton
Institute of Biomedical Research
College of Medical and Dental Sciences
University of Birmingham
Edgbaston
Birmingham B15 2TT
United Kingdom

Dear Dr. Frampton,

Thank you for submitting your manuscript entitled "Dependence on Myb expression is attenuated in myeloid leukaemia with N-terminal CEBPA mutations" to Life Science Alliance. The manuscript was assessed by the original reviewers again, whose comments are appended to this letter.

As you will, while reviewer #3 appreciates the introduced changes, reviewer #1 and #2 both think that their concerns have not been adequately addressed. They provide, however, constructive guidance on how to do so. Given the constructive input received, we would like to invite you to further revise your work. Importantly, such further revised version should

- include correct statistics and methods description as well as data representation (reviewer #1 general assessment and 'other points')
- acknowledge in the text that future work will have to show whether the findings are more generally applicable and applicable to human (reviewer #1)
- address the original comment 1 of reviewer #2 in a better way (compare effects of Myb KD in Cebpa mutant lines to the FMH9 cells; discuss the effects on apoptosis in a better way; discuss relevance to human disease)
- please also list 10 authors et al in your reference list
- please mention in the figure legends which statistical test has been used (you currently do this only in some legends)

Thank you for this interesting contribution to Life Science Alliance. We are looking forward to receiving your revised manuscript.

Sincerely,

- A letter addressing the reviewers' comments point by point.
- An editable version of the final text (.DOC or .DOCX) is needed for copyediting (no PDFs).
- High-resolution figure, supplementary figure and video files uploaded as individual files: See our detailed guidelines for preparing your production-ready images, <http://life-science-alliance.org/authorguide>
- Summary blurb (enter in submission system): A short text summarizing in a single sentence the study (max. 200 characters including spaces). This text is used in conjunction with the titles of papers, hence should be informative and complementary to the title and running title. It should describe the context and significance of the findings for a general readership; it should be written in the present tense and refer to the work in the third person. Author names should not be mentioned.

B. MANUSCRIPT ORGANIZATION AND FORMATTING:

Full guidelines are available on our Instructions for Authors page, <http://life-science-alliance.org/authorguide>

Reviewer #1 (Comments to the Authors (Required)):

In their revised manuscript, Volpe et al. have made a significant effort address the issues brought up with respect to providing statistics and reporting methodology that were initially brought up by this reviewer. However, some problems remain that are readily addressed by providing statistics as well as reporting the number of replicates within an experiment and in some cases the number of individual experiments. Also, the initial review pointed out that the work only reports results from 3 mouse cell lines and makes no attempt to extend the work to human samples. The manuscript is of clear interest and deserves publication but it is incumbent on the authors to acknowledge the need to extend the findings to more cell lines of the KL and LL mutations and determine if their findings apply more generally as well as to extend the work to human samples and determine if the findings apply to human AML.

Other points:

1. It remains unclear how the dye dilution assays were analyzed. The reader is not provided with the GMIs and the portions of each curve compared are not marked. There appear to be clear shifts in the KL (and maybe the LL) dye dilution curves at each time point. Most flow cytometry analytical software packages provide analysis of dye dilution data that determine precursor and responder frequencies as well as proliferative capacity. At a minimum, the GMIs of the regions that are being compared need to be provided in a bar graph with statistical variance along with a clear statement as to the number of independent experiments included in the graph. The portions of each curve being compared should be marked. Simply adding an "ns" in the box without clear indication of what it refers to is not adequate.

2. How the BrdU uptake/cell cycle analysis was carried out needs to be clarified. The Methods section states that cells were labeled with "7AAD or BrdU" per reference 29. How was BrdU uptake assigned to different stages of the cell cycle if the cells were not simultaneously assayed by cell labeling and flow cytometry? I am assuming that the authors meant that the cells were assessed for co-staining of DNA and BrdU by flow cytometry. These flow cytometry plots should be provided in the Supplemental data as they are important for interpreting changes or lack of changes in DNA synthesis and proliferation. Furthermore, by gating appropriately, sub-2n DNA content might provide further support for apoptosis.

3. The Annexin V experiments are difficult to interpret without inclusion of an intercalating dye and as the authors state in their rebuttal, they cannot distinguish between apoptotic and necrotic cell death by this method. Thus, labeling Figure 1G "apoptotic" cell death is not appropriate. Labeling as "Percent dead cells" or "Percent Annexin V+ cells" is appropriate. Other methods to support apoptotic cell death are available and are not difficult to carry out by flow cytometry such as PARP cleavage or using an anti-active caspase antibody. Alternatively, the use of ZVAD or other caspase inhibitors can be used to rescue cells from apoptotic cell death. Without better support for apoptosis the term cell death is more appropriate.

4. Changes in Bcl2 and Bim mRNA expression are not convincing, particularly since no statistics are provided to substantiate the small differences in expression that are detected. Also, the number of replicates in each qPCR run is not provided (2, 3, 4?) nor the number of independent experiments. If only 2 replicates are used statistics can be difficult. Protein blots would be more convincing since both Bim and Bcl-2 are subject to regulation by post-transcriptional mechanisms that can increase or decrease stability. However, statistically significant changes in mRNA expression would be

consistent with changes in Bcl-2 and Bim activity and provide some support for apoptotic cell death.

5. The Figure label "B" in Fig 1B is partially blocked by another figure component.

6. There are several places where "Myb" is used rather than c-Myb when talking about the protein. This can be confusing since the accepted symbol for the mouse c-Myb locus is italicized Myb.

Reviewer #2 (Comments to the Authors (Required)):

Review of REVISED VERSION of Volpe et al for Life Science Alliance

"Dependence on Myb expression is attenuated in myeloid leukaemia with N-terminal CEBPA mutations"

Comments for the authors

The authors have addressed many of my concerns in the revised version, including those about missing information raised by myself and by other reviewers. However some have not been comprehensively addressed. Below are my comments on the original version (reviewer), the authors' responses and my responses to those

Reviewer:

1. While the main conclusion is the most interesting part of the m/s, I have a number of concerns regarding the magnitude and significance of the effect seen. The "resistance" to MYB knockdown seen with the KL lines that represent the most common type of CEBPA-mutant AML is much less marked than that seen with the LL line that represents bi-allelic N-terminal mutant AML, which is rarely if ever seen in human disease. Indeed Fig 2 shows no resistance to Myb knockdown-induced differentiation in the case of the KL line. The magnitude of the proliferative effect is also hard to gauge since there is only a small increase in G0/G1 cells even in the control line (numbers are not given but it looks like ~ 58% to ~ 64%). THE CFSE data are a bit more convincing but even there the effect in the control cells is quite small at the time point examined. It may be useful to look at later time points. Moreover, there is a large increase in apoptosis seen in the LL and KL cells on Myb knockdown which isn't seen in the control. There is no examination or discussion of why this might be - this should be included. For example, is Bcl2, a known MYB target, down-regulated in these cell lines upon Myb knockdown?

Author response:

As commented by the reviewer, we have further examined the cell cycle changes between the different cell lines in the presence of reduced Myb levels and provided the percentage of cells in G0/G1 (Supplementary Figure S1B). In the case of the control line, an increase in G0/G1 from 51% to 74% with concomitant decrease in S-phase and G2/M phases was observed (Supplementary Figure S1C). Statistical analysis has also been included to reinforce the cell cycle changes observed in response to Myb KD in FMH9 cells (see Supplementary Figure S1C).

Regarding the apoptosis data, we have added in Supplementary Figure S2B a plot showing the differential regulation of Bcl2, a known Myb target and apoptosis regulator. This shows that Bcl2 is down regulated in response to Myb KD in KL cells only, in agreement with the strongest induction of apoptosis observed in those cells upon Myb down regulation. We have also added a plot showing the expression of Bcl2l11 (Bim), another regulator of apoptosis that has been shown to be anti-correlated with Myb in leukaemia. This has been added in the text under the subheading "Molecular

consequences of Myb manipulation carrying either wild type or mutant CEBPA" and relative references have been inserted. We hope that the reviewer would now find these data more convincing.

Reviewer response:

The additional data and statistics support the authors' conclusions that there is little effect of Myb KD on cell cycle progression in either LL or KL cells but an effect is seen in FMH9 cells. However the proliferation/cell count data of Fig 1D seems at odds with this as there is a decrease upon Myb KD for all 3 cell lines, albeit greatest for the control. The additional statistics here (significance thresholds indicated by asterisks) are for a comparison with the negative control siRNA. While this is OK as such, the real question is whether the effect of Myb siRNA seen with the Cebpa mutant lines is statistically different from that seen with the FMH9 cells - but this is not addressed.

This leads to the other point I made previously, ie, the effects on apoptosis. The revised m/s still doesn't really indicate whether the authors believe this could be responsible for the decreased cell numbers seen on Myb KD for the LL and KL lines in Fig 1D in the absence of a cell cycle effect. Moreover, while I appreciate the authors' addition of new data on Bcl2 and Bim expression, these don't really address why apoptosis is seen in the KL and LL lines, since the effect of Myb KD on expression of these genes is negligible, and unlikely to account for the observed level of apoptosis. Moreover I cannot agree that the difference in Bcl2 expression between LL and KL correlates with anything - it is far too small to be biologically meaningful. Moreover both lines show significant induction of apoptosis (~ 5- and ~10-fold for KL and LL, respectively). At the very least some comments in the Discussion on this are warranted.

Finally there is no response to my point that in human AML bi-allelic N-terminal mutations are rarely if ever seen ie whether the differentiation resistance of LL cells on Myb KD has any relevance to human disease.

Reviewer:

2. The gene expression studies in this system show a very different pattern for the LL cells of the KL cells which, as the authors say does suggest a difference in MYB dependency of many genes. How this is can be interpreted though is hard to know, because we have no idea about the relative chromatin occupancy of sites by the 2 mutant alleles. Indeed, this is something of a problem for the whole paper, because it relies on speculative inferences based on chromatin occupancy by the WT isoforms (from the second part of the paper).

Author response:

We agree with the reviewer that unravelling the binding dynamics of C/EBP β in WT and CEBPA mutant AML cell lines would have improved the manuscript tremendously and we understand the concerns raised by the reviewer as our data is based on chromatin occupancy by the WT isoform. To overcome this limitation, we sought to use previously published data that have been generated using FDCP1 expressing the p30 isoform only (which is essentially mimicking the expression pattern seen in the N-terminal mutant cells - i.e. LL), p42 only and have added data from a mutant version of the p42 isoform (namely K313KK), which disables the DNA binding domain, thus mimicking the pattern observed in C-terminal mutant cells. The expression of those isoforms, either WT or mutant, lead to an increase of C/EBP β expression that is similar to the expression pattern observed when comparing WT and CEBPA mutant AML cell lines used in this study. To provide a clearer explanation of our data, we have now added a comparison between these three isoforms in Figure 4A, 4C, 5A, 5C and 5D.

Reviewer response:

While as the authors acknowledge there is still much complexity to be unravelled, these additions do support the authors' conclusions.

Reviewer:

3. It is not clear how the KL and LL lines were derived - was it from published knock-in mice? Much more detail is needed.

Author response:

Indeed, those cells lines have been derived from published knock-in mice generated by the group of Prof Claus Nerlov (Kirstetter et al, 2008 Cancer Cell; Bereshchenko O et al, 2010, Cancer Cell). In response to the reviewer's comments we have added a brief explanation about the origin of the cells and establishment of the cell line together with the references to the original mouse models from which the cells have been derived and a reference of the study in which the cells lines have been generated. This information has been added in the "Cell lines" section of the Material and Methods.

Reviewer response:

OK now

Reviewer:

4. It is unclear how good a control FMH9 is for the KL and LL lines. How comparable are they with respect to lineage stage and potential? The data of Fig S1B suggest they may not be that well matched because of the large differences in gene expression in the absence of Myb knockdown. Similarly, the CHIP data on P42 and p30 occupancy are derived from yet another cell line - FDCP1.

Author response:

In response to the reviewer's comment we have added a phenotypic characterization of the cell lines used in this study. This information is provided in Supplementary Figure S1A.

Reviewer response:

The new information is welcome and indicates that the FMH9 control is similar with respect to these surface markers to LL cells, but differs somewhat from KL.

Reviewer:

5. There are a lot of other important details missing. For example, what are the data sets used in the second part of the m/s from? References to papers are provided, but GEO or similar accession numbers for the data actually used are essential. Also it is unclear where the "LSC Programme" dataset (Fig S2B) is from.

Author response:

We agree with the reviewer on the importance of providing GEO accession numbers for publicly available data. This has been added in the Supplementary Information in the section "Published CHIP-Seq data processing". CHIP data from the occupancy of p30 and p42 have been obtained upon request from the Grebien Lab. The reference about the LSC program has been added to the text (Ref 37, Gal, H et al. Gene expression profiles of AML derived stem cells; similarity to hematopoietic stem cells. *Leukemia* 20, 2147-2154, doi:10.1038/sj.leu.2404401 (2006)).

Reviewer response:

OK now

Reviewer:

6. A couple of more minor points: (i) The Discussion on p8 top paragraph cites a Ref 54, but the last reference in the reference list is # 42. (ii) GFi1 is actually a MYB target gene (Zhao et al Oncogene, 33:4442-44492 (2014)), it doesn't just "appear" to be positively regulated by MYB (p7).

Author response:

This mistake has been corrected according to the reviewer's comment and the correct reference (Zhao et al, 2014, Oncogene) has been added in the Discussion.

Reviewer response:

OK

Reviewer:

Overall, the most positive aspect of the paper is that it provides the first clear example of an AML type with reduced Myb dependency at least with regard to proliferation, and differentiation in the case of the LL line. Has some implications for human AMLs, since even KL cells showed some Myb independent proliferation. It would be good to see if this observation is replicated in human AML.

Author response:

We agree with the reviewer that the replication of our results in human cells would be of fundamental importance and would boost the value of our manuscript. However, we have not managed to obtain any access to primary human AML material from patients carrying CEBPA mutations.

Reviewer response:

This is unfortunate, and further underscores my concerns re direct relevance to human disease. However the m/s does provide potentially important insights into how CEBPA mutations might exert their effects in AML.

Reviewer:

The second part of the paper though is somewhat hard to follow. A major difficulty is that the first part of the paper deals with the effects of mutant CEBPA while the latter deals with WT. The analyses and interpretation of the previously-published data on p30 and p42 chromatin occupancy would benefit from consideration of the functional properties of p30 cf p42 proteins and mutants thereof in interpreting the results.

Author response:

The answer to this comment has already been provided in point 2.

Reviewer response:::

As above

Reviewer #3 (Comments to the Authors (Required)):

No further comments, I think the important comments raised by the referees have been adequately addressed.

Reviewer #1 (Comments to the Authors (Required)):

1. It remains unclear how the dye dilution assays were analyzed. The reader is not provided with the GMIs and the portions of each curve compared are not marked. There appear to be clear shifts in the KL (and maybe the LL) dye dilution curves at each time point. Most flow cytometry analytical software packages provide analysis of dye dilution data that determine precursor and responder frequencies as well as proliferative capacity. At a minimum, the GMIs of the regions that are being compared need to be provided in a bar graph with statistical variance along with a clear statement as to the number of independent experiments included in the graph. The portions of each curve being compared should be marked. Simply adding an "ns" in the box without clear indication of what it refers to is not adequate.

We apologize for not being clear enough with regard to the analysis of the CFSE proliferation assay presented in Figure 1E. To answer the reviewer, we have reanalyzed our data using Flowjo software and have obtained the geometric mean intensities (GMI) of CFSE fluorescence for every time point. In order to provide a clearer indication of the proliferation defect observed in FMH9 cells only, we have now presented our data as overlay histograms showing siNEG transfected cells with continuous lines and si*Myb* transfected cells with dotted line and we have compared the GMIs of siNEG and si*Myb* treated cells for every cell line at each time point using Student's t-test. The regions that are being compared are indicated by a color-matched bar on top of every peak together with statistical significance. We have also added a bar plot showing the average of GMIs for every cell line in FigS1B. The details about how many experiments have been performed and how the statistical analysis was performed have been added in the figure legend.

2. How the BrdU uptake/cell cycle analysis was carried out needs to be clarified. The Methods section states that cells were labeled with "7AAD or BrdU" per reference 29. How was BrdU uptake assigned to different stages of the cell cycle if the cells were not simultaneously assayed by cell labeling and flow cytometry? I am assuming that the authors meant that the cells were assessed for co-staining of DNA and BrdU by flow cytometry. These flow cytometry plots should be provided in the Supplemental data as they are important for interpreting changes or lack of changes in DNA synthesis and proliferation. Furthermore, by gating appropriately, sub-2n DNA content might provide further support for apoptosis.

We understand the confusion we have created. The Methods section should state that the cells were labelled with 7-AAD and BrdU. We have now rectified this. As the reviewer pointed out, BrdU/7-AAD co-labelling allowed for the assignment of different cell cycle phases. As requested by the reviewer, in Figure S2A we have added the flow cytometry plots from the BrdU incorporation assay to show how cells have been gated and the relative percentages of cells in every phase of the cell cycle to better appreciate the phenotypic change observed in response to *Myb* knockdown. For this analysis, cells were gated removing the "sub-2n population" for assigning specific cell cycle phases to "alive cells". Moreover, in Figure S2C we have added an overlay histogram showing the 7AAD fluorescence intensity of siNEG vs si*Myb* treated cells with which is possible to observe a sub2n population indicative of apoptotic/necrotic cells and a peak indicative of cell death. In

support of our conclusion, this is observed only in the CEBPA mutant lines but not in FMH9 cells. We hope that the reviewer finds these data more convincing.

3. The Annexin V experiments are difficult to interpret without inclusion of an intercalating dye and as the authors state in their rebuttal, they cannot distinguish between apoptotic and necrotic cell death by this method. Thus, labeling Figure 1G "apoptotic" cell death is not appropriate. Labeling as "Percent dead cells" or "Percent Annexin V+ cells" is appropriate. Other methods to support apoptotic cell death are available and are not difficult to carry out by flow cytometry such as PARP cleavage or using an anti-active caspase antibody. Alternatively, the use of ZVAD or other caspase inhibitors can be used to rescue cells from apoptotic cell death. Without better support for apoptosis the term cell death is more appropriate.

We thank the reviewer for his comment; we have now changed the label of the boxplots presented in figure 1G to "percentage of Annexin V+ cells" and we made changes in the text accordingly.

4. Changes in Bcl2 and Bim mRNA expression are not convincing, particularly since no statistics are provided to substantiate the small differences in expression that are detected. Also, the number of replicates in each qPCR run is not provided (2, 3, 4?) nor the number of independent experiments. If only 2 replicates are used statistics can be difficult. Protein blots would be more convincing since both Bim and Bcl-2 are subject to regulation by post-transcriptional mechanisms that can increase or decrease stability. However, statistically significant changes in mRNA expression would be consistent with changes in Bcl-2 and Bim activity and provide some support for apoptotic cell death.

We thank the reviewer for his comment. We have now performed quantitative RT-PCR analysis using 4 replicates to determine the consequences of Myb knock-down on the expression of both Bcl2 and Bim. This shows no evident changes in their expression in FMH9 cells, while a stark down-regulation of Bcl2 was observed in KL cells and an increase in Bim expression is seen in LL cells. A new panel showing these data has been added in Figure S3. The statistical significance has now been determined and described in the figure legend.

5. The Figure label "B" in Fig 1B is partially blocked by another figure component.

We thank the reviewer for pointing this out; this mistake has now been rectified.

6. There are several places where "Myb" is used rather than c-Myb when talking about the protein. This can be confusing since the accepted symbol for the mouse c-Myb locus is italicized Myb.

The accepted convention on nomenclature applied to MYB for gene/protein is *MYB*/*MYB* for human and *Myb*/*Myb* for mouse. We thus would like to keep this nomenclature in the revised version of the manuscript even though the reviewer raised some concern about this issue.

Reviewer #2 (Comments to the Authors (Required)):

Reviewer response:

The additional data and statistics support the authors' conclusions that there is little effect of Myb KD on cell cycle progression in either LL or KL cells but an effect is seen in FMH9 cells. However the proliferation/cell count data of Fig 1D seems at odds with this as there is a decrease upon Myb KD for all 3 cell lines, albeit greatest for the control. The additional statistics here (significance thresholds indicated by asterisks) are for a comparison with the negative control siRNA. While this is OK as such, the real question is whether the effect of Myb siRNA seen with the Cebpa mutant lines is statistically different from that seen with the FMH9 cells - but this is not addressed.

We thank the reviewer for this comment. To address this, we have calculated a ratio siNEG/siMYB at every time point and used this to compare the significance of the growth retardation defect between the three different cell lines. Using this approach, we have observed that indeed the growth defect observed in the CEBPA wild type cells is significantly more pronounced when compared to the CEBPA mutant lines. The statistical significance has been calculated using a Student's t-test. A bar plot representing this analysis has now been added in FigS1B and has been commented in the results section in the paragraph "Manipulation of Myb expression does not reverse the differentiation block in cells carrying biallelic N-terminal CEBPA mutations".

This leads to the other point I made previously, ie, the effects on apoptosis. The revised m/s still doesn't really indicate whether the authors believe this could be responsible for the decreased cell numbers seen on Myb KD for the LL and KL lines in Fig 1D in the absence of a cell cycle effect. Moreover, while I appreciate the authors' addition of new data on Bcl2 and Bim expression, these don't really address why apoptosis is seen in the KL and LL lines, since the effect of Myb KD on expression of these genes is negligible, and unlikely to account for the observed level of apoptosis. Moreover I cannot agree that the difference in Bcl2 expression between LL and KL correlates with anything - it is far too small to be biologically meaningful. Moreover both lines show significant induction of apoptosis (~ 5- and ~10-fold for KL and LL, respectively). At the very least some comments in the Discussion on this are warranted.

We believe in the apoptotic/necrotic phenotype observed in both CEBPA mutant cell lines that is not seen in CEBPA wild type cells; however, we also think that other biological pathways could be affected that lead to such a different response. For instance, the large cell death that is seen in LL cells could also be due to those cells undergoing other non-apoptotic fates, such as necroptosis, autophagic cell death, or pyroptosis in response to Myb knock-down. This would need to be investigated in further studies together with the relevance to the human disease. This comment has been added in the discussion.

Finally there is no response to my point that in human AML bi-allelic N-terminal mutations are rarely if ever seen ie whether the differentiation resistance of LL cells on Myb KD has any relevance to human disease.

We apologize to the reviewer for having accidentally omitted the answer to this comment. Indeed, the number of patients harbouring biallelic N-terminal mutations is much smaller compared to patients carrying C-ter/N-ter mutations, those being 80-90% of all CEBPA biallelic mutants. However, the exact percentage of N-ter/N-ter is not really known as most of the studies publicly available only report the presence of either monoallelic or biallelic mutations in the C/EBPa gene but fail to report whether those mutations are occurring at the C-terminus or N-terminus. There is only one study (Taskesen E et al, 2011) in which the percentage of both Cter/Nter and Nter/Nter mutants was assessed, the latter being roughly 6% of all patients with biallelic CEBPA, those in turn being roughly 6% of all AML patients within that dataset. There are many other datasets in which the percentage of biallelic CEBPA occurrences are almost 20%, thus potentially elevating the number of those patients. As such, we think that investigating how Nter/Nter mutant would respond to a candidate therapeutic approach is relevant to the human leukaemia.

Reviewer response:

While as the authors acknowledge there is still much complexity to be unravelled, these additions do support the authors' conclusions.

Reviewer:

3. It is not clear how the KL and LL lines were derived - was it from published knock-in mice? Much more detail is needed.

Author response:

Indeed, those cells lines have been derived from published knock-in mice generated by the group of Prof Claus Nerlov (Kirstetter et al, 2008 Cancer Cell; Bereshchenko O et al, 2010, Cancer Cell).

In response to the reviewer's comments we have added a brief explanation about the origin of the cells and establishment of the cell line together with the references to the original mouse models from which the cells have been derived and a reference of the study in which the cells lines have been generated. This information has been added in the "Cell lines" section of the Material and Methods.

Reviewer response:

OK now

Reviewer:

4. It is unclear how good a control FMH9 is for the KL and LL lines. How comparable are they with respect to lineage stage and potential? The data of Fig S1B suggest they may not be that well matched because of the large differences in gene expression in the absence of Myb knockdown. Similarly, the ChIP data on P42 and p30 occupancy are derived from yet another cell line - FDCP1.

Author response:

In response to the reviewer's comment we have added a phenotypic characterization of the cell lines used in this study. This information is provided in Supplementary Figure S1A.

Reviewer response:

The new information is welcome and indicates that the FMH9 control is similar with respect to these surface markers to LL cells, but differs somewhat from KL.

Reviewer:

5. There are a lot of other important details missing. For example, what are the data sets used in the second part of the m/s from? References to papers are provided, but GEO or similar accession numbers for the data actually used are essential. Also it is unclear where the "LSC Programme" dataset (Fig S2B) is from.

Author response:

We agree with the reviewer on the importance of providing GEO accession numbers for publicly available data. This has been added in the Supplementary Information in the section "Published CHIP-Seq data processing". CHIP data from the occupancy of p30 and p42 have been obtained upon request from the Grebien Lab. The reference about the LSC program has been added to the text (Ref 37, Gal, H et al. Gene expression profiles of AML derived stem cells; similarity to hematopoietic stem cells. *Leukemia* 20, 2147-2154, doi:10.1038/sj.leu.2404401 (2006)).

Reviewer response:

OK now

Reviewer:

6. A couple of more minor points: (i) The Discussion on p8 top paragraph cites a Ref 54, but the last reference in the reference list is # 42. (ii) GFi1 is actually a MYB target gene (Zhao et al *Oncogene*, 33:4442-44492 (2014)), it doesn't just "appear" to be positively regulated by MYB (p7).

Author response:

This mistake has been corrected according to the reviewer's comment and the correct reference (Zhao et al, 2014, *Oncogene*) has been added in the Discussion.

Reviewer response:

OK

Reviewer:

Overall, the most positive aspect of the paper is that it provides the first clear example of an AML type with reduced Myb dependency at least with regard to proliferation, and differentiation in the case of the LL line. Has some implications for human AMLs, since even KL cells showed some Myb independent proliferation. It would be good to see if this observation is replicated in human AML.

Author response:

We agree with the reviewer that the replication of our results in human cells would be of fundamental importance and would boost the value of our manuscript. However, we have

not managed to obtain any access to primary human AML material from patients carrying CEBPA mutations.

Reviewer response:

This is unfortunate, and further underscores my concerns re direct relevance to human disease. However the m/s does provide potentially important insights into how CEBPA mutations might exert their effects in AML.

Reviewer:

The second part of the paper though is somewhat hard to follow. A major difficulty is that the first part of the paper deals with the effects of mutant CEBPA while the latter deals with WT. The analyses and interpretation of the previously-published data on p30 and p42 chromatin occupancy would benefit from consideration of the functional properties of p30 cf p42 proteins and mutants thereof in interpreting the results.

Author response:

The answer to this comment has already been provided in point 2.

Reviewer response:

As above

Reviewer #3 (Comments to the Authors (Required)):

No further comments, I think the important comments raised by the referees have been adequately addressed.

February 28, 2019

RE: Life Science Alliance Manuscript #LSA-2018-00207-TRR

Prof. Jon Frampton
Institute of Biomedical Research
College of Medical and Dental Sciences
University of Birmingham
Edgbaston
Birmingham B15 2TT
United Kingdom

Dear Dr. Frampton,

Thank you for submitting your revised manuscript entitled "Dependence on Myb expression is attenuated in myeloid leukaemia with N-terminal CEBPA mutations". Reviewer #1 appreciates the introduced changes and supports the publication of your work. We would thus be happy to publish your paper in Life Science Alliance pending final revisions necessary to meet our formatting guidelines:

- please list 10 authors et al. in your reference list (currently often 1 author et al. listed)

A. FINAL FILES:

-- Summary blurb (enter in submission system): A short text summarizing in a single sentence the study (max. 200 characters including spaces). This text is used in conjunction with the titles of papers, hence should be informative and complementary to the title. It should describe the context and significance of the findings for a general readership; it should be written in the present tense

and refer to the work in the third person. Author names should not be mentioned.

B. MANUSCRIPT ORGANIZATION AND FORMATTING:

Sincerely,

Reviewer #1 (Comments to the Authors (Required)):

I am satisfied with the response of Volpe and colleagues to my last set of comments regarding their

manuscript.

March 7, 2019

RE: Life Science Alliance Manuscript #LSA-2018-00207-TRRR

Prof. Jon Frampton
Institute of Biomedical Research
College of Medical and Dental Sciences
University of Birmingham
Edgbaston
Birmingham B15 2TT
United Kingdom

Dear Dr. Frampton,

Thank you for submitting your Research Article entitled "Dependence on Myb expression is attenuated in myeloid leukaemia with N-terminal CEBPA mutations". It is a pleasure to let you know that your manuscript is now accepted for publication in Life Science Alliance. Congratulations on this interesting work.

*****IMPORTANT:** If you will be unreachable at any time, please provide us with the email address of an alternate author. Failure to respond to routine queries may lead to unavoidable delays in publication.*******

DISTRIBUTION OF MATERIALS:

Again, congratulations on a very nice paper. I hope you found the review process to be constructive and are pleased with how the manuscript was handled editorially. We look forward to future exciting

submissions from your lab.

Sincerely,
